
# How should we aggregate data? Methods accounting for the numerical distributions, with an assessment of aerosol optical depth

Andrew M. Sayer[1,2] and Kirk D. Knobelspiesse[2]

[1]GESTAR, Universities Space Research Association, Columbia, MD, USA
[2]NASA Goddard Space Flight Center, Greenbelt, MD, USA

**Correspondence:** Andrew M. Sayer (andrew.sayer@nasa.gov)

**Abstract.** Many applications of geophysical data - whether from surface observations, satellite retrievals, or model simulations - rely on aggregates produced at coarser spatial (e.g. degrees) and/or temporal (e.g. daily, monthly) resolution than the highest available from the technique. Almost all these aggregates report the arithmetic mean and standard deviation as summary statistics, which are what data users employ in their analyses. These statistics are most meaningful for Normally-distributed data; however, for some quantities, such as aerosol optical depth (AOD), it is well-known that distributions are on large scales closer to Lognormal, for which geometric mean and standard deviation would be more appropriate. This study presents a method to assess whether a given sample of data are more consistent with an underlying Normal or Lognormal distribution, using the Shapiro-Wilk test, and tests AOD frequency distributions on spatial scales of 1° and daily, monthly, and seasonal temporal scales. A broadly consistent picture is observed using Aerosol Robotic Network (AERONET), Multiangle Imaging Spectroradiometer (MISR), Moderate Resolution Imagining Spectroradiometer (MODIS), and Goddard Earth Observing System Version 5 Nature Run (G5NR) data. These data sets are complementary: AERONET has the highest AOD accuracy but is sparse; MISR and MODIS represent different satellite retrieval techniques and sampling; as a model simulation, G5NR is spatiotemporally complete. As time scales increase from days to months to seasons, data become increasingly more consistent with Lognormal than Normal distributions, and the differences between arithmetic and geometric mean AOD become larger, with geometric mean becoming systematically smaller. Assuming Normality systematically overstates both the typical level of AOD and its variability. There is considerable regional heterogeneity in the results: in low-AOD regions such as the open ocean and mountains, often the AOD difference is sufficiently small (<0.01) as to be unimportant for many applications, especially on daily timescales. However, in continental outflow regions and near source regions over land, and on monthly or seasonal time scales, the difference is frequently larger than the Global Climate Observation System (GCOS) goal uncertainty on a climate data record (the larger of 0.03 or 10 %). This is important because it shows the sensitivity to averaging method can and often does introduce systematic effects larger than the total goal GCOS uncertainty. Using three well-studied AERONET sites, the magnitude of estimated AOD trends is shown to be sensitive to the choice of arithmetic vs. geometric means, although the signs are consistent. The main recommendations from the study are that (1) the distribution of a geophysical quantity should be analysed in order to asses how best to aggregate it; (2) ideally AOD aggregates such as satellite level 3 products (but also ground-based data and model simulations) should report geometric mean or median rather than (or in addition to) arithmetic mean AOD; and (3) as this is unlikely in the short term due to the computational burden involved, users can calculate geomet-



ric mean monthly aggregates from widely-available daily mean data as a stopgap, as daily aggregates are less sensitive to the choice of aggregation scheme than those for monthly or seasonal aggregates. Further, distribution shapes can have implications for the validity of statistical metrics often used for comparison and evaluation of data sets. The methodology is not restricted to AOD and can be applied to other quantities.

## 1 Introduction

Geophysical data are obtained from a variety of data sources and model simulations across many disciplines in the Earth Sciences. As one example, aerosol optical depth (AOD) is often measured on the ground by Sun photometry (e.g. Giles et al., 2019), retrieved from passive (single- or multi- spectral, view, and polarisation state) or active (lidar) satellite observations (e.g.

Kokhanovsky and de Leeuw, 2009; Lenoble et al., 2013; Dubovik et al., 2019), and simulated by global models (e.g. Kinne et al., 2006). While each sensor or model has its own distinct spatial and temporal sampling characteristics, for applications in many research areas it is common to use aggregates represented by daily to seasonal averages and on length scales of order tens of km to several degrees. These are often somewhat coarser than the highest resolution available from a technique. For satellite retrievals, these daily or monthly aggregates are known as level 3 (L3) data. Level 2 (L2) data represent an instantaneous

snapshot, often along the orbit track at the native resolution of the sensor (or some multiple of it), and level 1 (L1) data consist of the geolocated satellite measured radiances which are used as inputs to L2 algorithms. Daily L3 data are constructed by aggregating L2 retrievals; monthly L3 data are typically constructed by aggregating daily L3, although in some cases have also been constructed from L2 directly, which gives different results if the contributing days have unequal sampling (Levy et al., 2009).

Reasons for preferring L3-type (i.e. aggregated) data for some applications over L2-type include the decreased storage and computational overhead, the fact that aggregates are typically reprojected onto a regular grid and so often more user-friendly, and a desire to have a data set with fewer gaps. Gaps can be caused by unfavourable retrieval conditions; for example, algorithms to retrieve atmospheric aerosol or surface reflective/emissive properties often require cloud-free, snow-free, and daytime scenes. Gaps also arise from the simple fact that surface/satellite observations do not observe every location all the

time. Unfortunately, sampling incompleteness adds an additional representivity error in comparisons; in some fields, such as aerosol remote sensing, this can be difficult to quantify and sometimes not negligible (Levy et al., 2009; Sayer et al., 2010; Colarco et al., 2014; Geogdzhayev et al., 2014; Schutgens et al., 2017). While global/regional model simulations are typically already on a fixed grid and spatiotemporally complete, the use of daily or monthly model averages likewise has the appeal of lower computational requirements and ease-of-use, particularly when comparing to an incomplete ground-based or satellite

product.





While the principles of uncertainty propagation in remote sensing are well established (Povey and Grainger, 2015; Merchant et al., 2017), until recently comparatively little effort (relative to L1 and L2 development) has been put into determining the most meaningful ways to construct L3 data and assess their uncertainties. This is despite the wide use of these data products in research. One notable exception is sea surface temperature, for which comprehensive estimates of multiple components of L3

uncertainties have been developed (Kennedy, 2014; Bulgin et al., 2016a, b). Implicit in the calculation of summary statistics such as mean and standard deviation in a L3-type data set (or model average) is the assumption that the points aggregated belong to some local population, such that the calculation of summary statistics is meaningful for describing the state of the Earth. Use of binned data is another option, although analyses using binned aggregates are generally less common than those using averages. One fundamental aspect of this is the question of how to average the data, i.e. which distribution's summary

statistics provide the most useful and meaningful metrics to report. No simple distribution is likely to provide a perfect fit to any observational data set, so the relevant problem is in finding an approximate distribution sufficient for a particular application. Choice of mean (and often additionally standard deviation), as is most common in many fields (including AOD), takes as given that the Normal distribution (which is described in terms of these two parameters) is an appropriate distribution to summarise this population. For a given mean $\bar{\tau}_n$ and standard deviation $\sigma_n$ of AOD, the Normal frequency distribution $P(\tau) \sim \mathcal{N}(\bar{\tau}_n, \sigma_n^2)$

is given by

$$P(\tau) = \frac{1}{N}\frac{\mathrm{d}N}{\mathrm{d}\tau} = \frac{N}{\sqrt{2\pi}\sigma_n}\exp\left[-\frac{1}{2}\left(\frac{\tau - \bar{\tau}_n}{\sigma_n}\right)^2\right], \tag{1}$$

where $N$ is a normalisation constant (the total number of data points). As this is symmetric about $\bar{\tau}_N$, this mean value is also the distribution's median and mode.

This assumption runs counter to the fact that AOD at a given location tends not to be Normally distributed, which has been

indicated in the literature for at least 50 years. Writing in terms of aerosol-induced turbidity (directly proportional to AOD), Flowers et al. (1969) presented measurements at $500\,\mathrm{nm}$, collected through the early 1960s using sun photometers designed by Volz (1959), as part of an observation network of several dozen sites across the United States of America (USA). Note that this was but one of several networks observing atmospheric turbidity (sometimes separating aerosols from other contributions, sometimes not) with various types of instrument through the 20th century. Holben et al. (2001) reviews others, with the earliest

being bolometer measurements in Washington, District of Columbia (DC), USA, beginning in 1902 (Roosen et al., 1973). Instrumentation and data processing (e.g. calibration, data collection/reporting, cloud screening) methods limit the accuracy and use of some of these earlier records; Forgan et al. (1993) provide a thorough discussion. Nevertheless, Flowers et al. (1969) found (their Figure 4) cumulative distribution functions consistent with Lognormal distributions, i.e., Normal when the data are represented in log space; analogous to Equation 1, the Lognormal frequency distribution $P(\log_{10}\tau) \sim \mathcal{L}(\bar{\tau}_l, \sigma_l^2)$ is given

by

$$P(\log_{10}\tau) = \frac{1}{N}\frac{\mathrm{d}N}{\mathrm{d}\log_{10}\tau} = \frac{N}{\sqrt{2\pi}\sigma_l}\exp\left[-\frac{1}{2}\left(\frac{\log_{10}\tau - \log_{10}\bar{\tau}_l}{\sigma_l}\right)^2\right]. \tag{2}$$

Here $\bar{\tau}_l$, $\sigma_l$ are the geometric mean and geometric standard deviation of AOD respectively. Base 10 logarithm is used here for numerical convenience. For easier comparison between the two distribution forms, in this study $\bar{\tau}_n$ (i.e. arithmetic mean)





and $\bar{\tau}_l$ (i.e. geometric mean) are represented and will be discussed in absolute, rather than logarithmic, units. Note that due to the additive properties of logarithms $\bar{\tau}_l$ is equivalent whether calculated as the geometric mean of $\tau$, or the arithmetic mean of $\log_{10} \tau$, i.e.:

$$\bar{\tau}_l = \left( \prod_{i=1}^{N} \tau_i \right)^{\frac{1}{N}} = 10^{\frac{1}{N} \sum_{i=1}^{N} (\log_{10} \tau_i)} \tag{3}$$

Geometric standard deviation $\sigma_l$ is the standard deviation of log-transformed data ($\log_{10} \tau$). Because of this, unlike arithmeric standard deviation, it is a multiplicative rather than additive factor (cf. Kirkwood, 1979), i.e. the central one standard deviation of the data are encompassed by the range $10^{\log_{10}(\bar{\tau}_l - \sigma_l)}$ to $10^{\log_{10}(\bar{\tau}_l + \sigma_l)}$ (multiplicative), implying an asymmetric range when expressed in absolute (non-logarithmic) units, compared to $\bar{\tau}_n \pm \sigma_n$ (additive) for an arithmetic mean.

     Note that Equation 2 is often expressed in terms of $\mathrm{d}N/\mathrm{d}\tau$ rather than $\mathrm{d}N/\mathrm{d}\log_{10} \tau$ (i.e. linear rather than logarithmic

ordinate). In this case, using the chain rule and properties of logarithms, the relation between the two formulations is given by

$$\frac{\mathrm{d}N}{\mathrm{d}\tau} = \frac{\mathrm{d}N}{\mathrm{d}\log_{10} \tau} \frac{\mathrm{d}\log_{10} \tau}{\mathrm{d}\tau} = \frac{\mathrm{d}N}{\mathrm{d}\log_{10} \tau} \frac{1}{\ln(10)\tau} \tag{4}$$

where $\ln(10)$ denotes the natural logarithm of 10, $\approx 2.30$. Some further relations between Normal and Lognormal distribution parameters are given by Table 1 of O'Neill et al. (2000).

     Other studies published around this time (e.g. Ahlquist and Charlson, 1967; Volz, 1970; Volz and Sheehan, 1971; Rangara-

jan, 1972) reported AOD measurements in other parts of the world. These analyses were more concerned with estimating the value and distribution (which turned out to be close to Normal) of its wavelength-dependence, via the Ångström exponent $\alpha$ (Ångström, 1929), than that of AOD. This was of interest both for visibility applications and because $\alpha$ was often used to estimate one of the parameters in the aerosol particle size distribution model of Junge (1955, 1963), which was used widely at the time. Intriguingly, one implication of Lognormally-distributed AOD is that $\alpha$ should be Normally distributed (if the data

belong to a single population). This arises from the definition of $\alpha$,

$$\alpha = -\frac{\mathrm{d}\log(\tau(\lambda))}{\mathrm{d}\log(\lambda)} \approx -\frac{\log \frac{\tau_{\lambda_1}}{\tau_{\lambda_2}}}{\log \frac{\lambda_1}{\lambda_2}} = -\frac{\log \tau_{\lambda_1} - \log \tau_{\lambda_2}}{\log \lambda_1 - \log \lambda_2}, \tag{5}$$

for AOD ($\tau$) at some wavelength $\lambda$, approximated in these studies using bispectral AOD measurements at wavelengths $\lambda_1$, $\lambda_2$. Due (again) to the additive properties of algorithms, $\alpha$ as the log-ratio of two Lognormal distributions is equivalent to the difference of two Normally-distributed quantities (even when they are correlated, as is the case for AOD), which is itself

Normally distributed. If AOD were Normally distributed, then (because it is a positive-definite quantity) in low-AOD conditions $\alpha$ would exhibit significant skew and possibly multiple modes (in high-AOD conditions $\alpha$ might appear close to Normal but with incorrect kurtosis). Hence, the close-to-Normality of the $\alpha$ distributions presented in some of those studies, given the fairly low-AOD conditions, support (although are not alone unambiguous evidence for) Lognormally-distributed AOD populations. One caveat is that $\alpha$ distributions can exhibit false skew dependent on the magnitude and spectral correlation of the uncertainties

in $\tau(\lambda)$ (Wagner and Silva, 2008). Note that Equation 5 is insensitive to choice of logarithmic base.





Daily and monthly averages of extinction at multiple locations presented by Roosen et al. (1973) also show skewed distributions associated with Lognormality, although frequency distributions are not directly shown. Several years later, Malm et al. (1977) and King et al. (1980) presented spectral AOD measurements from the opposite ends (Page and Tucson, respectively) of Arizona, USA. They realised that it was most appropriate to represent the resulting frequency distributions with logarith-

mic (geometric), rather than arithmetic, averages and standard deviations. More recent work has taken advantage of the great increase in data quality, volume, and coverage possible from better instrumentation and computational power. O'Neill et al. (2000) showed that AOD derived from sun photometer measurements at a variety of individual Aerosol Robotic Network (AERONET) sites spread around the world tends to have frequency distributions which statistically resemble a Lognormal distribution to a much stronger degree than Normal. All these previous studies were of data aggregated over time summarised

at individual locations; around the same time, providing an early satellite example, Ignatov and Stowe (2000) found approximately Lognormal AOD (and Normal $\alpha$) in aerosol retrievals over ocean scenes. This indicated that Lognormal tendencies might be found in AOD data also aggregated spatially, as opposed to just temporally. Similar skewed distributions were reported by Smirnov et al. (2011) for ship-based Sun photometer AOD observations taken on cruises. Maps of retrieved or simulated AOD, and scatter density plots in satellite validation studies, show a similar pattern (e.g. Kinne et al., 2006; Remer et al., 2008;

Sayer et al., 2012): a large cluster of points at a comparatively low AOD, with a rapidly-decreasing number of points as AOD increases, corresponding to locations and times affected by severe smoke, dust storms, or pollution episodes.

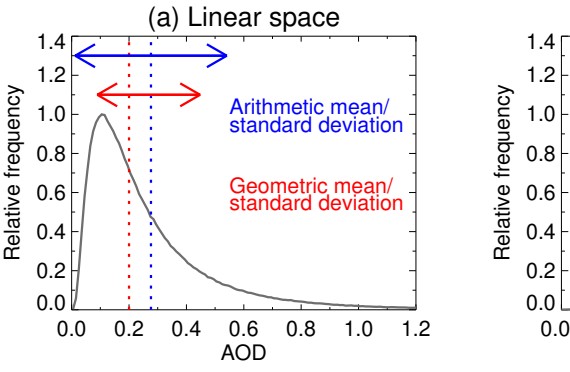
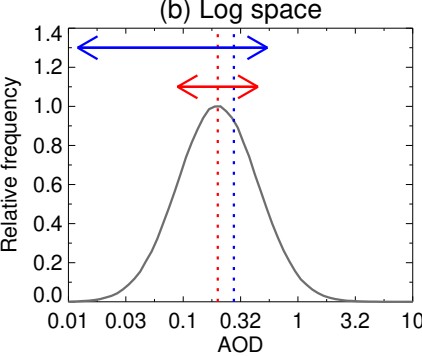

**Figure 1.** Synthetic frequency distributions for lognormally-distributed AOD with a mean of 0.2 and geometric standard deviation 0.35, $\mathcal{L}(0.2, 0.35^2)$, shown on (a) linear and (b) logarithmic AOD axes. Vertical red and blue dashed lines represent geometric and arithmetic mean values, respectively. Horizontal red and blue arrows indicate the range of geometric and arithmetic mean ± one standard deviation.

Due to this asymmetry Normal statistics (i.e. arithmetic mean $\bar{\tau}_n$ and standard deviation $\sigma_n$) will overstate the typical level of AOD observed and its variability, implying in some cases unphysical negative AOD. This is illustrated in Figure 1, which compares arithmetic and geometric statistics for a synthetic AOD distribution $\mathcal{L}(0.2, 0.35^2)$, similar to that of many locations

across the United States and Europe (e.g. O'Neill et al., 2000). The central one standard deviation ($1\sigma$) about the mean, which corresponds to an AOD range of 0.09-0.45 (i.e. $\log_{10}(\bar{\tau}_l) \pm \sigma_l$ in log space) when calculated using the geometric mean and standard deviation, correctly encompasses approximately 68.4 % of the data. In contrast calculating arithmetic mean and





standard deviation gives 0.28 (i.e. overstating the typical AOD) and 0.27; the resulting $1\sigma$ range ($\bar{\tau}_n \pm \sigma_n$, 0.01-0.55) includes 89.2 % of the data (i.e. overstating the variability). Figure 1(b) reveals the symmetry of the distribution when shown in log space. Thus, representing a Lognormally-distributed quantity using Normal-appropriate statistics has systematic quantitative implications for the interpretation of the data.

Lognormal distributions are common across quantities in the natural sciences, and tend to arise when the underlying phenomenon is governed partly by multiplicative (rather than additive) factors; Limpert et al. (2001) provide general, and Hinds (1999) and Anderson et al. (2003) aerosol-specific, examples and discussion. For aerosols these factors may include, for example, changes in emissions or removal (e.g. onset of fires/soil fragmentation on the one hand, or precipitation on the other) which are not linear in effect. As well as AOD, aerosol particle size distributions may be represented sufficiently well by combinations of lognormal modes (Dubovik et al., 2002) that this is common practice in satellite retrieval algorithms. Kok (2011a, b) presented a theoretical model of dust emission based on fragmentation theory and lognormal size distributions, which agreed with measurements better than existing parametrisations in global models. Similar behaviour is found for many other remotely-sensed quantities; for example, Campbell (1995) assessed Lognormality on large scales for oceanic chlorophyll, water-leaving radiance, and photosynthetic yield, while cloud optical depth (COD) is also known to be distributed approximately Lognormally (King et al., 2013), and some of the most widely-used cloud classification schemes (introduced by Rossow and Schiffer, 1999) are based on joint histograms of (roughly-lognormal) COD against cloud-top pressure.

Recent efforts by other researchers have helped to understand spatial and temporal scales in AOD variations and their potential effects on data aggregates. Anderson et al. (2003) used surface-level aerosol scattering and column AOD and found autocorrelation could remain high on scales of tens to several hundreds of km, and time scales of days to weeks. Noting that study, Kovacs (2006) assessed validation statistics of Moderate Resolution Imaging Spectroradiometer (MODIS) AOD against AERONET as a function of the distance of satellite retrievals from AERONET sites. The level of agreement showed site-specific drop-offs with distance, with generally less variability over ocean sites which were less likely to be influenced by local sources. Alexandrov et al. (2004) used a network of shadowband radiometers across the Southern Great Plains in the USA to perform an energy spectrum analysis on AOD variations. They observed a scale break at length scales around 12-15 km (interestingly, slightly larger than many spaceborne L2 AOD products), below which the structure function of AOD variations showed one exponent, and above which they showed another, corresponding to regimes where variations were dominated by 3D and 2D turbulence, respectively. Using field campaign observations and satellite retrievals over the southeastern USA, Kaku et al. (2018) note that correlation lengths can differ for surface-level vs. column aerosol loading. These studies of correlation structure are important for defining suitable scales for a population to be aggregated, and for describing how the error characteristics of such aggregates might vary spatially and temporally.

Several studies have sought to assess representation uncertainty in L3-type aggregates; Sayer et al. (2010) examined how the completeness of sampling of satellite AOD retrievals within model grid cells affected the level of agreement between data sets. Li et al. (2016) assessed how representative long-term AERONET sites are on satellite L3 spatial scales and monthly time scales, as part of a larger body of work to characterise and reduce the uncertainty in multi-sensor monthly mean AOD records. From a perspective of comparing global model grid cells to point measurements, Schutgens et al. (2016) assessed to





what extent representativeness errors caused by coarse model grid size could be decreased by temporal averaging. They found that AOD uncertainties could be decreased to a greater extent than other aerosol properties, but that such errors were often still larger than desirable. Schutgens et al. (2017) then attempted to estimate representation uncertainties on ground- or satellite-type

aerosol data aggregates on different spatiotemporal scales. They found that spatiotemporal collocation was important and, as in the prior study, representation errors could still be significant in some cases, such as when near aerosol point sources or in complex terrain. Alexandrov et al. (2016) proposed describing the logarithm of AOD in terms of Gaussian structure functions (accounting for aerosol loading, variance, and autocorrelation), and presented a comparison between MODIS retrievals with global circulation model simulations represented in this way. Povey and Grainger (2019) aggregated satellite AOD retrievals

on a monthly basis (i.e. L2 to monthly L3 directly) and represented the results in terms of sums of lognormal modes. They found that doing so both highlighted regions of significant variability, and aided in identifying systematic differences between data sets.

This analysis aims to complement these other recent studies, building most directly on O'Neill et al. (2000), as a further step towards a more robust calculation and use of AOD aggregates in ground-based, satellite, and model-simulation studies.

While the example application is to AOD, the framework introduced is applicable more generally to other (geophysical or not) data aggregates. The central questions to be addressed are: on commonly-used spatial and temporal scales, does a Normal or Lognormal distribution better represent AOD frequency distributions? When and where does the choice matter? When and where might neither distribution be adequate? And what are the implications for L3 data and related analyses if a Lognormal representation is used instead? Section 2 describes the data and methodology employed. Section 3 presents the results of the

analysis, and Section 4 discusses the implications of the findings for the creation and use of aggregated AOD data or model simulations.

## 2    Definitions, methodology, and data

### 2.1    Data and model simulations used

This analysis uses ground-based observations from AERONET, together with satellite retrievals from the Multiangle Imag-

ing Spectroradiometer (MISR) and MODIS instruments, and model simulations from the Goddard Earth Observing System (GEOS) Version 5 Nature Run (G5NR). All of these have different spatiotemporal sampling techniques and associated uncertainties in their estimates of AOD. Considering a diverse set of data sources such as this provides a more comprehensive picture of the frequency distributions of AOD than would be obtained from only a single data type. It allows the strengths of individual techniques to be used, while helping to avoid erroneous conclusions stemming from limitations of individual techniques. The

data sources are described below.



### 2.1.1 AERONET

AERONET provides aerosol (and water vapour) data from Sun photometer measurements, obtained with standardised acquisition, calibration, and processing protocols. This analysis uses the latest version 3 direct-Sun level 2 (cloud-screened, post-deployment calibrated, and quality-assured) AERONET AOD data. Note 'level 2' in AERONET terminology refers to quality-assurance level, regardless of temporal aggregation level, as opposed to satellite level 2 which refers to instantaneous data only. Version 3 includes improvements to sensor characterisation, site geolocation accuracy, and cloud-aerosol discrimination (Giles et al., 2019), particularly in the detection of stable optically-thin cirrus cloud layers, and rapidly-evolving fine-mode aerosol plumes. Here, all direct-Sun observations from all sites (1185 at the time of writing) from the start of 1993 to the end of 2018 are used. Measurement cadence depends on the instrument model used and can be adjusted dependent on the desired mix of scan types, but for direct-Sun observations is typically every 5-15 minutes in cloud-free skies during daylight hours.

All instruments deployed as part of AERONET provide AOD at 440, 675, 870, and $1020\,\mathrm{nm}$ at a minimum; the majority include additional channels between 340 and $1600\,\mathrm{nm}$, with $500\,\mathrm{nm}$ being a common addition. In this analysis, AERONET AOD are interpolated spectrally to $550\,\mathrm{nm}$, as this is a common reference wavelength for many satellite data products and model simulations, although the conclusions do not change if other wavelengths are used instead. Hereafter, mentions of AOD without a specified wavelength refer to AOD at $550\,\mathrm{nm}$. This is performed with a least-squares fit of all available AERONET AODs within the 440-870 nm wavelength range (typically 4, more for some configurations) to a quadratic polynomial,

$$\log(\tau_\lambda) = a_0 + a_1\log(\lambda) + a_2\log(\lambda)^2, \tag{6}$$

where coefficients $a_0$, $a_1$, $a_2$ are calculated on a point-by-point basis. This quadratic formulation is more robust to calibration problems in individual channels. It also reflects the fact that the relationship between $\log(\tau)$ and $\log(\lambda)$ is not linear but shows curvature dependent on fine mode particle size (Eck et al., 1999; Schuster et al., 2006). When more than two wavelengths are available, this is a more realistic description of the spectral derivative of AOD than the bispectral approximation in Equation 5. The uncertainty on AERONET midvisible AOD is $\sim 0.01$ (Eck et al., 1999), somewhat smaller than typical uncertainties on satellite retrievals or model simulations.

### 2.1.2 MISR

The latest version 23 of MISR L2 data provides AOD at $558\,\mathrm{nm}$, over land and ocean, with a horizontal pixel size of $4.4\,\mathrm{km}$; use of 558 rather than 550 nm has negligible impact on the analysis here. The instrument includes 9 cameras with a maximum swath width around $400\,\mathrm{km}$, although the edges of the scan are not covered by all cameras and so retrievals are provided over a slightly narrower swath. This provides repeat views of a given scene roughly once per week at tropical latitudes and once every three days at high latitudes. MISR flies on the Sun-synchronous Terra platform, providing data from early 2000, with a 10:30 a.m. local solar Equatorial crossing time. Separate processing algorithms are applied over land and dark water; version 23 updates and initial evaluation are provided by Garay et al. (2017) and Witek et al. (2018). These have not yet been validated on a global basis, but are expected to reduce some high biases seen over low-AOD water scenes, and low biases seen over high-AOD scenes, reported in validation analyses of previous data versions (e.g. Kahn et al., 2010).





This analysis uses five years (2004-2008) of L2 data, corresponding to around half a million retrievals per day (after accounting for unfavourable retrieval conditions). The choice of record length is a balance between robustness of the analyses and storage/processing concerns; one year of the MISR L2 product (MIL2ASAE) corresponds to approximately 170 GB. As an order-of-magnitude estimate, assuming (on average) a revisit time of five days and half the data being unsuitable for retrieval due to e.g. cloudiness, approximately 200 views of a given point on the Earth would be expected over a five-year period. While this would show considerable spatial variation, qualitatively it is expected to be sufficient as it is well-known from observations and modeling that the main features of the global aerosol system are systematic and repeat year-to-year (e.g. d'Almeida et al., 1991; Holben et al., 2001; Kinne et al., 2006; Remer et al., 2008). Recently, Lee et al. (2018) used MISR (version 22) and MODIS retrievals to assess how many years of data were required (on both an annual and a seasonal basis) for a calculated climatology to converge to within an AOD of ±0.01. They found that over much of the open ocean and many land regions 5 or fewer years were sufficient, although for some aerosol source regions even the full MISR record (17 years at the time) was insufficient. This does not directly answer the question of what record length is necessary for the present analysis, although does suggest that except for near strong source regions the results should sample sufficient interannual variability to be only weakly sensitive to the specific time period chosen.

### 2.1.3 MODIS

The MODIS instruments fly on the Terra and Aqua platforms; L2 data from the latest Collection 6.1 (C61) from Aqua (launched 2002) are used here, for the same five-year period as the MISR analysis. MODIS Aqua is thought to have slightly better radiometric performance than Terra (e.g. Lyapustin et al., 2014). Additionally, the Aqua orbit has a 1:30 p.m. local solar Equatorial crossing time, so this provides a higher degree of sampling independence from the MISR retrievals than if MODIS Terra were used. The MODIS Atmospheres aerosol product used here (MYD04) includes retrievals from two Dark Target (DT) algorithms, for pixels identified as water and vegetated land respectively, plus the Deep Blue (DB) algorithm. The C61 DT land algorithm is similar to that of the previous Collection 6 (C6, Levy et al., 2013), but implements an updated surface reflectance model, detailed in Gupta et al. (2016), to reduce a systematic positive bias of DT over urban surfaces. The C61 DB data include numerous small updates to surface/aerosol models and cloud/quality assurance (QA) tests to reduce known error sources (Hsu et al., 2019; Sayer et al., 2019). All three algorithms also benefit from sensor calibration updates. Since C6, MODIS retrievals have included a QA-filtered merged data set combining DB and DT retrievals to increase spatial coverage. The C6 merging algorithm is described by Sayer et al. (2014), and essentially uses the water DT algorithm for water scenes and picks from or averages the DB and DT algorithms dependent on surface type over land. The same merging logic is applied in creation of the C61 merged product, which is used here. Note that the DT land algorithm permits retrieval of AOD down to -0.05, although negative AOD is unphysical; here, zero or negative AOD values are set to 0.0001 instead (as logarithms are only defined for positive values). The results of this analysis are negligbly sensitive to the choice of AOD floor threshold.

MODIS' 2,330 km swath results in near-global daily observations in the tropics, and once or twice-daily observations at higher latitudes. Retrievals are provided at 10 km nominal horizontal pixel size at the sub-satellite point. Towards the edge of the scan, the scan geometries and Earth's curvature cause a 'bow-tie distortion' where pixels become larger, and consecutive





scans begin to overlap (Xiong et al., 2006). This distortion at the edge of the swath is about a factor of two in the along-track

and five in the across-track direction (i.e. tenfold increase in pixel area), and overlap is close to 100 %, which has consequences

for AOD retrieval characteristics (Sayer et al., 2015). For about half of the swath, however, the areal expansion is less than a

factor of two compared to the nominal $10\,\mathrm{km} \times 10\,\mathrm{km}$ pixel size.

### 2.1.4 GEOS-5 Nature Run

The G5NR is a global $7\,\mathrm{km}$ non-hydrostatic mesoscale simulation based on the Ganymed version of GEOS-5 (Putman et al.,

2014). The aerosol component is described and evaluated by Castellanos et al. (2018). This is a two-year (May 2005-2007)

simulation, and while some factors (e.g. volcanic and biomass burning emission sources) were prescribed, meterology was not.

Hence, the G5NR is not a direct simulation of that specific historical period (and should not be compared one-to-one against

real observations from that period), but designed to provide a realistic and representative simulation of the Earth system from

which synthetic observations could be generated for e.g. observation system development.

Aerosol output fields are provided on a $30\,\mathrm{min}$ timestep on a $0.0625°$ regular latitude/longitude grid. This includes column

AOD contributed by organic carbon, black carbon, dust, sea salt, and sulfate, and following the recommendations of Castellanos

et al. (2018), scaling factors (their Table 3) are applied to these component AODs before summing to get the total AOD. These

scaling factors bring the G5NR component AODs in line with a climatology from the Modern Era Retrospective analysis for

Research and Applications Aerosol Reanalysis (MERRAero). MERRAero was a long-term reanalysis which assimilated

MODIS-based AOD; its aerosol component is evaluted by e.g. Buchard et al. (2015). Data from the simulated year 2006 only

are used; Castellanos et al. (2018) noted that G5NR aerosol fields were initialised to zero, and so did not use the initial six

months of the simulation to ensure that equilibrium had been reached. The final six months of the simulation are also discarded

here to ensure that each calendar month has equal representation in the analysis. As this leaves only four available seasons,

G5NR output is analysed on only a daily and monthly basis.

### 2.2 The Shapiro-Wilk test and its application

Shapiro and Wilk (1965) present a method to test whether a sample of data are consistent with draws from a Normally-

distributed population. Their derivation included some empirical comparisons to other tests, and they found it to have some

advantages over those techniques. Yap and Sim (2011) performed Monte Carlo simulations of various distributions to assess

eight different Normality tests, and found that the Shapiro-Wilk (SW) test has greatest statistical power in most circumstances.

The SW test computes the squared discrepancy between the quantiles of the sample with those expected from random samples

from a Normally-distributed population. Implementations are available in many software packages and languages. The test

statistic $W$ for a sample $x$ is defined

$$W = \frac{\left( \sum_{i=1}^{N} a_i x_{(i)} \right)^2}{\sum_{i=1}^{N} \left( x_i - \bar{x} \right)^2} \qquad (7)$$





where $x_{(i)}$ indicates the $i$th smallest sample member (known as the $i$th order statistic of $x$), and $a_i$ are weighting coefficients calculated from the expected values ($m$) and covariance ($V$) of order statistics from Normally-distributed data:

$$(a_1, \ldots, a_N) = \frac{m^\mathrm{T} V^{-1}}{(m^\mathrm{T} V^{-1} V^{-1} m)^{1/2}} \tag{8}$$

The test requires $N \geq 3$, and $W$ can take values between 0 and 1. Sarhan and Greenberg (1956) (their Table 1), later corrected in Sarhan and Greenberg (1969), provide $V$ for $N \leq 20$; Shapiro and Wilk (1965) provide approximate calculations to larger sample sizes, and Royston (1982, 1992) provide approximate $m$ and $V$ up to $N = 5000$. Relevant sample sizes in the present study are up to several hundred points. As the Normal distribution is symmetrical about its mean, the coefficients $a_i$ are

symmetrical; e.g. $a = (-0.643, -0.281, -0.088, 0.088, 0.281, 0.643)$ for $N = 6$. Note the sign of the elements of $a_i$ is not relevant due to the square in the numerator of Equation 7. The coefficients are larger for the outer elements of $a_i$, corresponding to the tails of the data sample (i.e. the outer order statistics $x_{(i)}$). The numerator of Equation 7 thus represents a tail-waited sum of squares, while the denominator a sum of squared deviations from the sample mean $\bar{x}$. For Normally-distributed samples these increase around the same rate, such that $W$ is close to 1; for non-Normal data the denominator increases more rapidly

such that $W$ becomes closer to 0.

Royston (1992) provide a normalisation for $W$ in order to estimate a $p$-value for the result, i.e. the probability that a $W$ score at least as extreme would be observed under the Null hypothesis of the sample being drawn from a Normally-distributed population. The Royston (1992) extension for large $N$ and normalisation are used here. Note that the equivalent test for Lognormality is simply $W$ calculated using the logarithm of the data, i.e. here $\log_{10} \tau$ rather than $\tau$. A high $p$-value indicates

consistency with draws from a Normal (or Lognormal) distribution. One important point to note is that this test only tests the degree departure from Normality: it does not test the importance of that depature. As with any test like this, the power (i.e. efficacy at detecting a given departure) is a function of sample size. Thus for large sample sizes it is easier to detect a discrepancy from Normally-distributed data, even if the discrepancy is trivial. Both of these points should be borne in mind when interpreting the results.

The SW test is employed here as follows. First, spatial distributions of AOD are assessed by aggregating the MISR, MODIS, and G5NR data from their native spatial resolutions to $1°$ (as this is a common spatial scale for L3 AOD products and model output) and applying the SW test, without any aggregation in time. The resulting aggregates thus have a daily timestep for MISR and MODIS (considering all orbits from a given calendar day), and $30\,\mathrm{min}$ for G5NR. Next, temporal variations of AOD within a day are assessed by aggregating the AERONET and (previously spatially-aggregated) G5NR data on a daily

basis and applying the SW test to each site/grid cell. Aggregating G5NR first in space and then in time is more similar to the way polar-orbiting L3 aggregates sample the global aerosol system (as each L2 product is essentially a near-instantaneous snapshot), although the results are not significantly different if G5NR is analysed first in time and then in space. Finally, the resulting daily aggregates from all data sets are aggregated to monthly and seasonal time steps and the SW test applied to each site/grid cell. Seasons are defined December-January-February (DJF), March-April-May (MAM), June-July-August

(JJA), September-October-November (SON). Note the monthly/seasonal calculations use daily $\bar{\tau}_l$ as a basis, although results differ negligibly if $\bar{\tau}_n$ is used instead.





In each case, at least three data points are required for an aggregate to be considered valid; this is the minimum required for the SW test calculation, and also the minimum number of observations for AERONET or MODIS standard processing to report a daily average value, and the minimum number of days for MODIS products to report a monthly average value. The SW $p$-value is computed for distributions of $\tau$ and $\log_{10}\tau$, with resulting $p$-values denoted $p(\tau)$ and $p(\log_{10}\tau)$ respectively, and the results fall into one of four possible categories:

1. $|\bar{\tau}_l - \bar{\tau}_n| \leq \tau_t$. In this case the choice of Normal or Lognormal summary statistics may be considered unimportant, as the resulting arithmetic/geometric averages are similar. The threshold $\tau_t$ is taken as 0.01, which is the typical uncertainty on AERONET midvisible AOD (Eck et al., 1999), and thus represents a reasonable lower bound on achievable uncertainty on average AOD from models/observations at the present time. It is also similar to the thresholds for AOD accuracy over land ($\pm 0.016$) and ocean ($\pm 0.011$) estimated by Chylek et al. (2003) to be necessary to be able to constrain the aerosol direct radiative effect to $\pm 0.5\,\mathrm{Wm}^{-2}$.

2. $|\bar{\tau}_l - \bar{\tau}_n| > \tau_t$, but both $p(\tau) < p_t$ and $p(\log_{10}\tau) < p_t$. In this case both tests return a smaller $p$-value than some threshold $p_t$, indicating evidence of detectable deviation from both Normal and Lognormal distributions at this significance level. Here $p_t$ is taken as 0.001; if the underlying AOD data really were perfectly Normally or Lognormally distributed (and if the distinction was important) then approximately 0.1 % of data would be expected to fall into this category. However, in reality it is expected that the true distributions are neither of these, and additionally measurement/model errors may distort the observed distributions, leading to more points within this category. Since the $p$-value is not informative about the magnitude of a deviation from Normality/Lognormality, the additional criterion $|\bar{\tau}_l - \bar{\tau}_n| > \tau_t$ is included as it indicates that the magnitude of the AOD difference is sufficiently large that it might be important for some scientific applications (i.e. both statistically and scientifically relevant). Note that the analysis here is only weakly sensitive to the choice of $p_t$.

3. $p(\tau) > p(\log_{10}\tau)$, $p(\tau) > p_t$, and $|\bar{\tau}_l - \bar{\tau}_n| > \tau_t$. Here the data are more consistent with draws from a Normal than a Lognormal distribution, the data are reasonably consistent with a Normal distribution, and the difference between arithmetic and geometric mean AOD is not negligible. In this case use of Normal summary statistics is more appropriate.

4. $p(\tau) < p(\log_{10}\tau)$, $p(\log_{10}\tau) > p_t$, and $|\bar{\tau}_l - \bar{\tau}_n| > \tau_t$. The converse of category 3, here the data are best represented by Lognormal summary statistics.

The results will be interpreted in terms of relative frequencies of these four categories, as it is important to realise that the idiosyncrasies in real-world data complicate the estimation and calculation of $p$-values. For example, the ideal case of independent random samples from the true population cannot be achieved due to correlated errors in observations or simulations, and non-random sampling in space and/or time. SW (or other tests) cannot say whether or not the data are Normally/Lognormally distributed for any given instance, but instead only help say to what extent the two distributions are reasonable, useful approximations on the whole. In cases of small sample sizes the statistical power of the test may remain small; if the test results for a given area are essentially noise, then similar frequences of Normality and Lognormality might be expected. The best that can be done is to keep in mind the limitations of the data, and the statistical tests, in the interpretation of the analysis.





# 3 SW test categorisation results

## 3.1 Spatial and temporal variation within a day

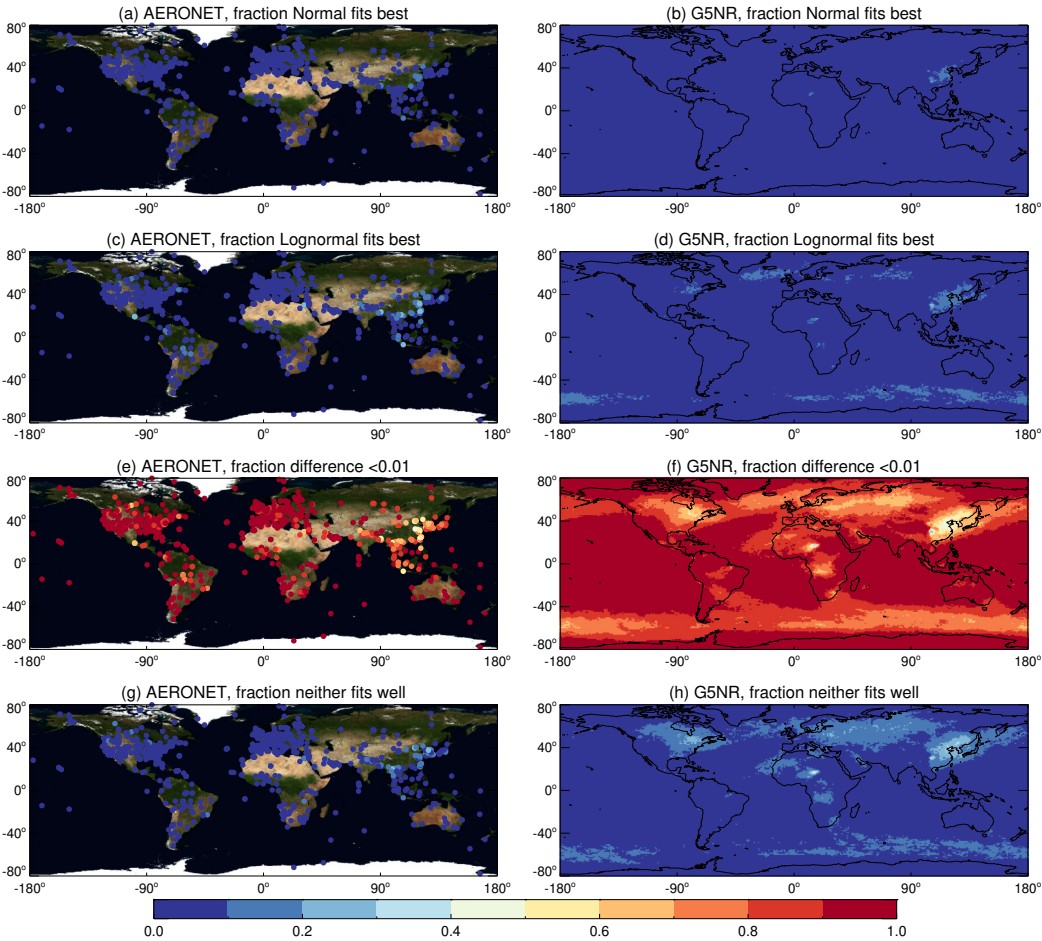

**Figure 2.** Fraction of data falling into each of the four categories of Shapiro-Wilk test results, for AOD distributions aggregated temporally over a day. Columns show (left) AERONET and (right) G5NR data. From top-bottom, rows indicate the fraction where the data are more consistent with a Normal distribution and $|\bar{\tau}_l - \bar{\tau}_n| > 0.01$; fraction more consistent with a Lognormal distribution and $|\bar{\tau}_l - \bar{\tau}_n| > 0.01$; fraction where $|\bar{\tau}_l - \bar{\tau}_n| \leq 0.01$; and fraction where $|\bar{\tau}_l - \bar{\tau}_n| > 0.01$ and the data show large discrepancies from frequencies expected by both Normal and Lognormal distributions. For AERONET, at least 50 days are required for a site to be considered valid.

Figures 2 and 3 respectively show the categorisation results for temporal (from AERONET and G5NR) and spatial (from MISR, MODIS, and G5NR) frequency distributions of AOD on daily scales. As a summary, Table 1 shows the global mean fractions of data in each category; note that these are the mean of each valid AERONET site/grid cell (i.e. all sufficiently-sampled areas are treated equally). Sites/grid cells require at least 50 valid days with data to be included in these statistics. As





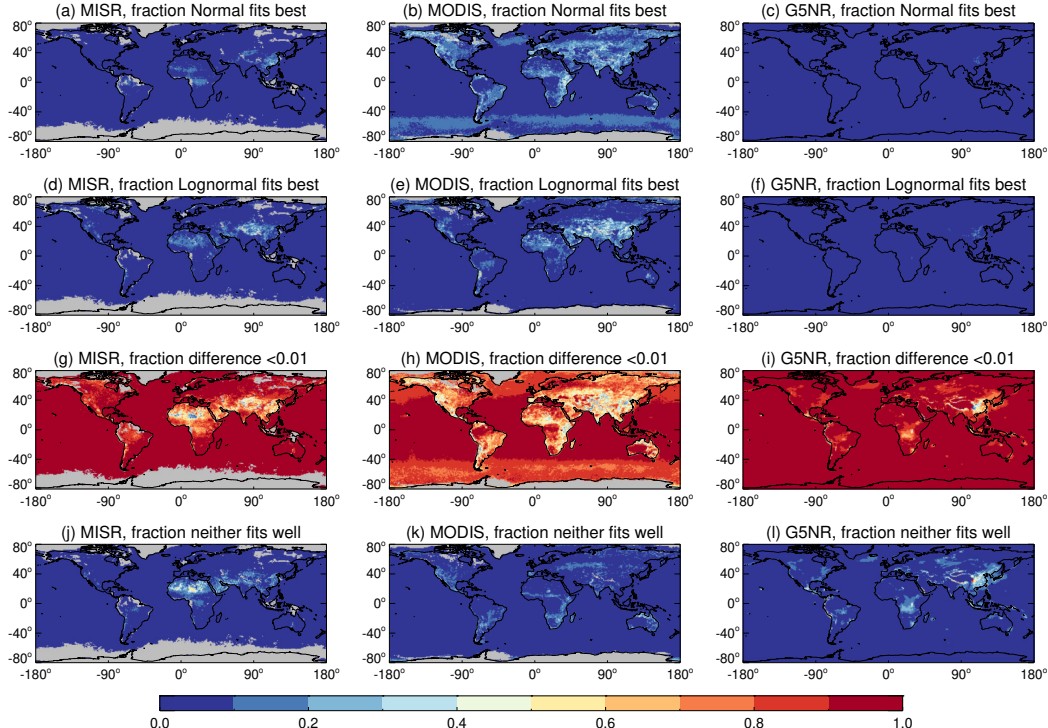

**Figure 3.** As Figure 2, except for AOD distributions aggregated spatially over a day from full resolution to $1°$. Columns show (left) MISR, (middle) MODIS, and (right) G5NR data. At least 50 contributing days are required for a grid cell to be valid; grid cells with insufficient data are shaded in grey.

the spatial sampling between the data sets is quite different (Figures 2, 3) the results from the different data sets are not expected to match, but reading the table left-right gives a sense for how the categorisations change on the different scales assessed. The following general conclusions can be drawn about variability relevant to daily aggregation:

10  1. Patterns shown between Figures 2 and 3 are similar, i.e. daily AOD frequency distributions tend to have similar shapes whether for temporal aggregation over a day (as from AERONET or model output) or spatial aggregation on scales of $1°$ (as from polar-orbiting satellites). This establishes that it is reasonable to aggregate spatial and temporal data on a daily basis in a similar way.

2. In areas of low to moderate AOD, including the global oceans, mountains, and fairly clean continental regions, for a strong majority (typically 80 % or more) of days the difference between arithmetic and geometric mean AOD ($|\bar{\tau}_l - \bar{\tau}_n|$) is smaller than 0.01. In these circumstances, calculating an arithmetic mean when the underlying distribution is Lognormal (or vice-versa) introduces an error smaller than 0.01.

5  3. In southern and eastern Asia and parts of North Africa, where the AOD is often high, the difference between arithmetic and geometric mean is more frequently (up to around half the time) larger than 0.01. This implies greater sensitivity to



**Table 1.** Mean fraction of data falling into the four categories of SW test results.

| Category/data set | Daily spatial | Daily temporal | Monthly temporal | Seasonal temporal |
|---|---|---|---|---|
| Difference $\|\bar{\tau}_l - \bar{\tau}_n\| \leq 0.01$ | | | | |
| AERONET | - | 0.892 | 0.436 | 0.315 |
| MISR | 0.942 | - | 0.647 | 0.503 |
| MODIS | 0.843 | - | 0.335 | 0.177 |
| G5NR | 0.961 | 0.916 | 0.679 | - |
| $\|\bar{\tau}_l - \bar{\tau}_n\| > 0.01$ and more consistent with Normal distribution | | | | |
| AERONET | - | 0.022 | 0.094 | 0.058 |
| MISR | 0.015 | - | 0.113 | 0.109 |
| MODIS | 0.074 | - | 0.272 | 0.290 |
| G5NR | 0.001 | 0.013 | 0.038 | - |
| $\|\bar{\tau}_l - \bar{\tau}_n\| > 0.01$ and more consistent with Lognormal distribution | | | | |
| AERONET | - | 0.043 | 0.465 | 0.594 |
| MISR | 0.022 | - | 0.239 | 0.385 |
| MODIS | 0.049 | - | 0.385 | 0.423 |
| G5NR | 0.003 | 0.025 | 0.275 | - |
| $\|\bar{\tau}_l - \bar{\tau}_n\| > 0.01$ and inconsistent with both distributions | | | | |
| AERONET | - | 0.043 | 0.006 | 0.033 |
| MISR | 0.021 | - | 0.0003 | 0.004 |
| MODIS | 0.034 | - | 0.007 | 0.111 |
| G5NR | 0.034 | 0.045 | 0.008 | - |

the choice of averaging method. For these cases, Lognormality tends to be a better representation of the distributions than Normality, although for a non-negligible fraction of the data neither distribution shape provides a good fit.

Figure 4 provides brief examples of AOD distributions falling into three of the SW test categorisations, for AERONET AOD data collected within a single day. As will be shown later, differences between Normal and Lognormal distributions become more pronounced at longer timescales than in these examples. The case for Midway Island (in the Pacific Ocean), a location dominated by low-AOD maritime conditions (Smirnov et al., 2003), shows a case where the arithmetic and geometric mean AOD are both around 0.055 and thus choice of summary statistic is likely unimportant for most applications (although note that $p(\log_{10}\tau) > p(\tau)$, indicating greater consistency with a Lognormal distribution). The case for Moscow (Russia) taken from a period of extreme wildfires during summer 2010, characterised by intense smoke (Chubarova et al., 2012). Here, the data are more consistent with a Normal distribution than Lognormal, and $\|\bar{\tau}_l - \bar{\tau}_n\| = 0.05$. This example ($N$=21) illustrates some difficulties in purely visual inference about distribution shape when histograms are sparse; the median number





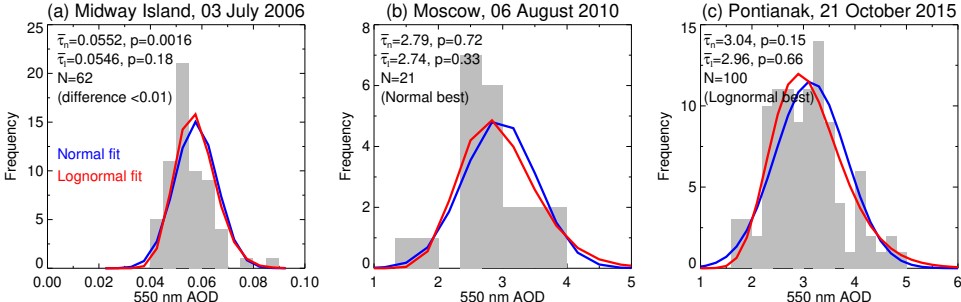

**Figure 4.** Histograms (grey) of 550 nm AOD observed at three AERONET sites on individual dates (given in panel titles), corresponding to different SW test classification results. Arithmetic and geometric mean AOD ($\bar{\tau}_n, \bar{\tau}_l$ respectively), $p$-values for the SW test for the respective distributions, number of points, and category are also given for each case. Bin sizes are site-dependent. Normal and Lognormal fits to each histogram are shown in blue and red, respectively.

of points contributing to a single day of AERONET data globally was 25. In this case the data are more consistent with a Normal distribution due to a closer match toward the tails of the distribution, but the data are consistent with both Normal and Lognormal distributions under most relevant significance levels. The final case is from Pontianak (Borneo, Indonesia) during an intense period of biomass burning in 2015, an event analysed in detail by Eck et al. (2019). For this date $|\bar{\tau}_l - \bar{\tau}_n| = 0.08$ and the data show greater consistency with Lognormality, due to the skew of the distribution.

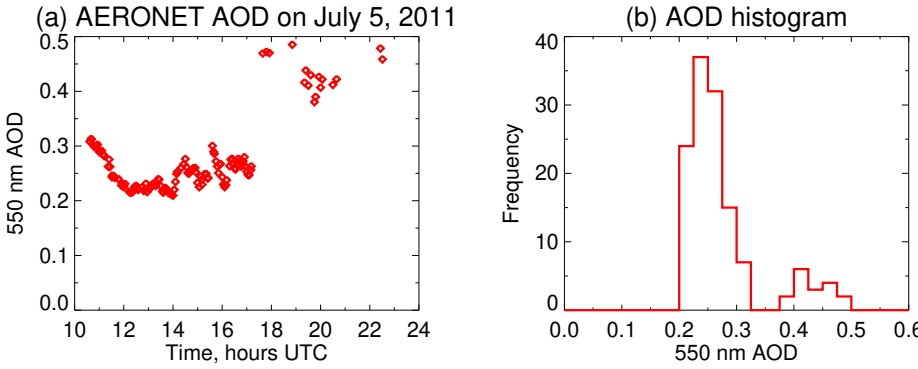

**Figure 5.** (a) Time series and (b) histogram (bin size 0.025) of 550 nm AOD observed at the Essex, Maryland AERONET site on 5 July 2011.

Days where neither Normal nor Lognormal distributions provide a good fit to AOD observations are commonly those where multiple regimes are present within a grid cell or during a day. Figure 5 presents a case study from the AERONET site in Essex, Maryland, USA on 5 July 2011. This day was previously analysed by Eck et al. (2014), as a case of rapid AOD enhancement following the development of a cumulus cloud field just after 17:00 Universal Time Coordinated (UTC), near solar noon (Figure 5a). The Sun photometer was operated with a 3 min sampling cadence, with 132 points in total throughout the day. For the day





as a whole $p(\tau) \ll p_t$ and $p(\log_{10} \tau) \ll p_t$, i.e. the data are inconsistent with both Normal and Lognormal distributions, revealed by the bimodality of the histogram in Figure 5b. However, for the 111 points before 17:00 UTC $p(\tau) = 0.002$ and $p(\log_{10} \tau) = 0.009$, and for the 14 after 18:00 UTC $p(\tau) = 0.38$ and $p(\log_{10} \tau) = 0.54$, in both cases indicating stronger evidence for Lognormality than Normality. By combining various data sets and lines of evidence, Eck et al. (2014) attribute enhancements like these to a combination of humidification and new particle formation rather than cloud contamination in the direct-Sun data, so there is physical reasoning for this bimodality. In situations like this the multimodal-fitting approach of Povey and Grainger (2019) would give a more complete representation of the aerosol field than presenting single-distribution summary statistics.

Note also that the near-universal choice of aggregating daily on a UTC calendar day basis, rather in terms of local solar time, can further complicate matters for locations far from the meridian. For example, AERONET sites in eastern Asia, Australasia, and the western Americas often contain data from midnight to mid-morning UTC, with a long gap, and then late evening to midnight UTC. The break in the middle is due to local nighttime, during which no data are collected; i.e. observations from a single UTC day can contain data from two local days. If something happens during this gap to affect the AOD distribution, which is often the case due to the diurnal variations or meteorology, this will naturally increase the chances of multimodality. Thus, something as basic as the definition of the day to aggregate to can affect the inferred AOD distribution shape. This could be contributing to some of the cases where neither distribution fits (Figures 2, 3) in these parts of the world. This affects all the data sets.

While similar, the patterns in Figures 2 and 3 are not identical between data sets. G5NR is the only data set which enables both spatial and temporal aggregation on a daily basis. Here, both aggregates show, for example, small differences between $\bar{\tau}_l$ and $\bar{\tau}_n$ over much of the global oceans and a higher frequency of large differences over southern and eastern Asia. However, the spatial aggregates also show areas of large difference (fit well by neither Normal nor Lognormal distributions) for grid cells with strong elevation variations such as along the edges of the Himalayas or Andes Mountains, while the temporal aggregates do not. If the bulk of the aerosol here is low-lying, then this naturally leads to another case of multiple populations within a grid cell. This is not seen to the same extent in the satellite retrievals here, although they are known to under-sample (due to misinterpreting spatial heterogeneity of the scene for cloud cover) and sometimes have retrieval artefacts which could distort the distributions (Zelazowski et al., 2011; Sayer et al., 2014; Loría-Salazar et al., 2016). In these cases moving to a finer spatial scale might be useful to provide summary metrics for these populations separately, i.e. $1°$ might be too coarse. The aforementioned retrieval artefacts might also explain some of the discrepancies between MODIS and other results in other mountainous areas such as western North America, Europe, and the Horn of Africa.

G5NR temporal aggregates also show increased incidence of Lognormality and of neither distribution fitting well in the Southern Ocean, while G5NR spatial aggregates do not; this implies diurnal cycles which affect the aerosol field here coherently on scales larger than $1°$. A similar feature, with 10-20 % occurrence of Normally-distributed AOD in the Southern Ocean, is seen in the MODIS results. MODIS retrievals are known to report higher AOD here than other data sets (including active sensors, Sun photometry, and data sets with stricter cloud screening); Toth et al. (2013) attributed much of this to a combination of cloud contamination and retrieval assumptions of surface wind speed (which affect surface brightness). This latter factor was addressed in more recent MODIS data versions (Levy et al., 2013) compared to those used by Toth et al. (2013), although the





35  enhanced AOD remains, implying that cloud contamination is still a factor. A similar enhancement was seen in older version of the MISR data product, but largely removed in the latest version used here (Witek et al., 2018). This implies that the occasional Normality seen in MODIS daily AOD in the Southern Ocean is likely to be an artefact of biases in the AOD retrievals. MODIS and MISR also report Normal or Lognormal AOD distributions each up to about 30 % of the time over various North African and Central Asian deserts, while G5NR does not. Unfortunately, the remoteness of many of these areas means that AERONET

5  has few sites in them. It is therefore hard to resolve the differences between the various data sets.

**Table 2.** Fractional SW test assigment of spatial AOD variation with a day from selected AERONET DRAGON-like deployments and clustered sites.

| Deployment location | IOP(s) | Maximum sites* | Number of days | Fraction in each category | | | |
|---|---|---|---|---|---|---|---|
| | | | | Normal | Lognormal | Difference $\leq 0.01$ | Neither fits |
| *Field campaign and similar deployments* | | | | | | | |
| Greater Washington, DC, USA | Jun-Aug 2007, Jul 2011 | 41 | 2627 | 0.057 | 0.086 | 0.855 | 0.004 |
| Osaka, Japan | Mar-Jun 2012 | 8 | 122 | 0.197 | 0.057 | 0.746 | 0.00 |
| Seoul, Korea | Mar-Jun 2012, May-Jun 2016 | 8 | 565 | 0.250 | 0.317 | 0.432 | 0.009 |
| Penang, Malaysia | Sep 2012 | 8 | 66 | 0.152 | 0.227 | 0.621 | 0.00 |
| Singapore | Sep 2012 | 5 | 50 | 0.100 | 0.260 | 0.640 | 0.00 |
| San Joaquin Valley, California, USA | Jan-Feb 2013 | 15 | 81 | 0.037 | 0.123 | 0.840 | 0.00 |
| Houston/Galveston, Texas, USA | Aug 2013 | 16 | 134 | 0.090 | 0.0448 | 0.858 | 0.007 |
| Colorado, USA | Jul 2014 | 15 | 1888 | 0.093 | 0.038 | 0.868 | 0.001 |
| Henties Bay, Namibia | Aug-Sep 2016 | 6 | 32 | 0.00 | 0.063 | 0.938 | 0.00 |
| *Locations with AERONET sites clustered within $\sim 100\,\mathrm{km}$* | | | | | | | |
| Abu Dhabi, United Arab Emirates | - | 3 | 35 | 0.114 | 0.057 | 0.829 | 0.00 |
| Beijing, China | - | 3 | 889 | 0.282 | 0.334 | 0.384 | 0.001 |
| New York, USA | - | 3 | 215 | 0.033 | 0.065 | 0.902 | 0.00 |
| Sierra Nevada, Spain | - | 3 | 88 | 0.091 | 0.193 | 0.716 | 0.00 |
| Taipei, Taiwan | - | 4 | 301 | 0.312 | 0.329 | 0.359 | 0.00 |
| Tenerife | - | 4 | 1574 | 0.370 | 0.174 | 0.456 | 0.001 |
| Western Provence, France | - | 3 | 380 | 0.026 | 0.058 | 0.916 | 0.00 |

*Maximum number of sites providing data on a single day; may be less than number of sites deployed in total.





AERONET also provides some opportunities to study the spatial distribution of AOD on horizontal scales of tens to around $100\,\mathrm{km}$, similar to L3/global climate model resolution. These are mostly commonly in so-called Distributed Regional Aerosol Gridded Observation Networks (DRAGONs) of up to several dozen sites, as detailed by Holben et al. (2018), deployed during Intensive Operating Period(s) (IOPs) of field campaigns. Some DRAGON deployments have been in areas also containing sev-

eral long-term AERONET sites (e.g. those around Washington, DC, USA), enabling spatial characterisation (to a lesser extent) outside these IOPs. Further, a few areas have had 3 or more AERONET sites deployed simultaneously within $\sim 100\,\mathrm{km}$ of each other; often (but not always) this overlap was temporary as one site replaced another. Table 2 shows the categorisation resulting from applying the SW and AOD difference threshold tests on daily geometric mean AOD for each of these field campaign deployments or groups of spatially-clusterd AERONET sites. More details of the DRAGON deployments are available in Holben

et al. (2018), and the AERONET webpage (https://aeronet.gsfc.nasa.gov) provides additional background information and the locations of other clustered sites. Categorisation results are broadly in line with Figures 2 and 3, and Table 1, in that typically the most common finding is that the difference between daily arithmetic and geometric mean AOD is smaller than 0.01. For the 10 field campaign deployment regions listed in Table 2, Lognormality is more commonly observed than Normality in six of them for the days when the resulting difference in AOD is at least 0.01. For the seven clusters of sites outside of field campaigns

(which have fewer, i.e. 3-4 sites total), Lognormality is more common in five. While this is consistent with the picture from the larger-scale analysis, it is also important to recall that these deployments are typically short in time (often weeks to months) and tend to be around major metropolitan areas. As a result the frequencies in Table 2 might not be extensible to longer time periods at these locations, or other environments.

### 3.2 Temporal variation on monthly and seasonal scales

Maps of categorisation of monthly and seasonal AOD aggregates, in both cases from daily AOD, are shown in Figures 6 and 7 respectively. Global-average fractions are again shown in Table 1. Monthly satellite/AERONET composites require at least 16 contributing months to be considered valid, and seasonal at least 8 contributing seasons; for the satellites, using five years of data, a maximum of 60 months or 15 seasons are possible. Increasing these thresholds removes some shorter-term AERONET sites, and satellite retrievals at high latitudes and some tropical locations, where retrieval coverage is limited. As only one year

of G5NR data is used, the monthly analysis is performed but seasonal analysis is not. Moving from daily to monthly aggregates in Table 1, the overall tendency is for AOD differences to become larger (i.e. the fraction within the category $|\bar{\tau}_l - \bar{\tau}_n| \leq 0.01$ decreases), and the distributions increasingly favour Lognormality over Normality. Going from monthly to seasonal, the trend is more pronounced, both in absolute fraction of data (Table 1) and in the spatial distributions (Figure 7). As in the daily data, some features are broadly consistent between the data sets:

1. Unlike daily aggregation, for monthly or seasonal aggregation the difference between arithmetic and geometric means is frequently more than 0.01. Thus, monthly/seasonal aggregates are more sensitive to the choice of averaging method. This implies generally larger variability on time scales of months/seasons than of spatial variability within a day, which

**Figure 6.** As Figure 2, except for AOD distributions aggregated temporally from daily to monthly. Columns show (from left to right) AERONET, MISR, MODIS, and G5NR data. Except for G5NR, at least 16 contributing months are required for an AERONET site or grid cell to be valid; grid cells with insufficient data are shaded in grey.



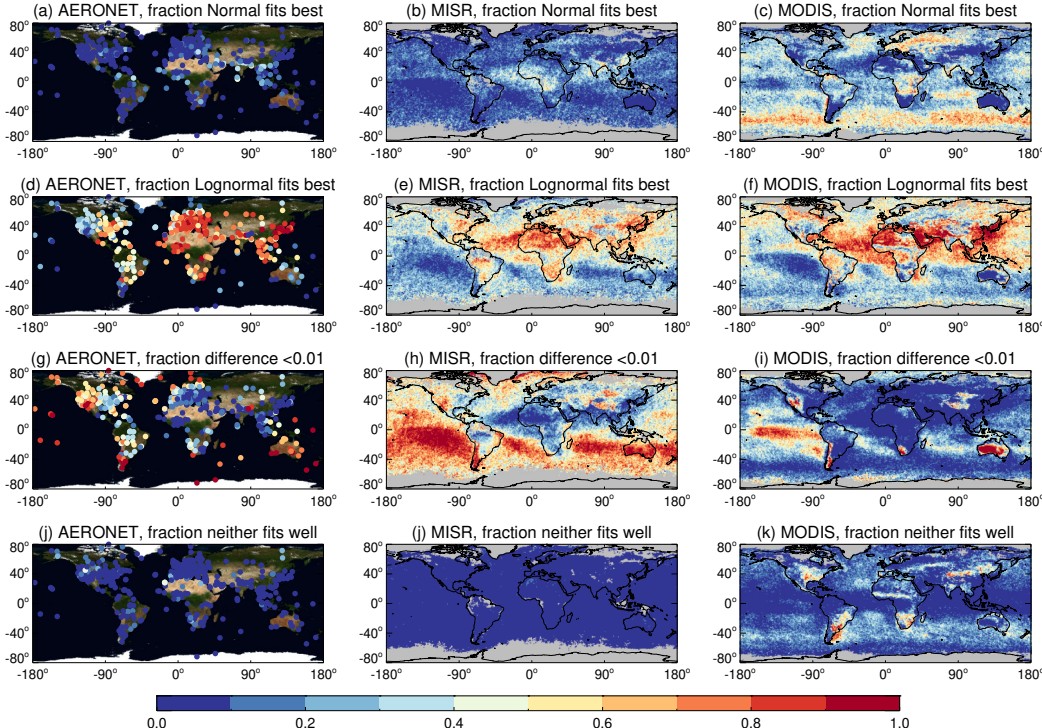

**Figure 7.** As Figure 2, except for AOD distributions aggregated temporally from daily to seasonal. Columns show (left) AERONET, (middle) MISR, and (right) MODIS data. At least 8 contributing seasons are required for an AERONET site or grid cell to be valid; grid cells with insufficient data are shaded in grey.

is consistent with previous work (e.g. Schutgens et al., 2017) that has established that temporal colocation on daily, rather than monthly, time scales is important to reduce sampling-related differences between AOD data sets.

2. The exception to the above is very clean areas: parts of the open ocean, Australasia, and mountainous/remote continental areas, that are outside of aerosol transport paths. Here, a plurality (but not always a majority) of the time the AOD difference remains less than 0.01.

3. Downwind of major aerosol source regions, over both and land and ocean, all data sets tend to report higher consistency more frequently with Lognormal than with Normal distributions.

Some of the differences between the data sets identified in the daily analysis, such as the southern ocean AOD in MODIS, are still present in the monthly and seasonal analyses. While patterns are often consistent, differences in magnitudes of each category may be driven in part by sampling differences, which are more pronounced at these scales. Of up to 31 days contributing to a month and 92 to a season, AERONET/MODIS often sample ~10-25 and 20-70, respectively (dependent on cloud cover and polar night) while MISR (due to its narrower swath) often samples only 3-7 days per month and 5-15 per season. Dependent on the temporal scales of aerosol system change, these differences may be important. This limited sampling accounts





for the sparser MISR coverage at high latitudes in Figure 6. Monthly and seasonal data are not affected by the same potential

15    'definition of day' issues as identified for daily composites. Seasonal aggregates may, however, be influenced by definition of

seasons, and in some parts of the world (e.g. South and East Asia, due to their Summer monsoons at various points from May to

September; Kang et al., 1999) other definitions than the canonical DJF, MAM, JJA, SON used here may be more appropriate.

## 4    Implications and recommendations

### 4.1    Magnitude of arithmetic vs. geometric mean AOD differences

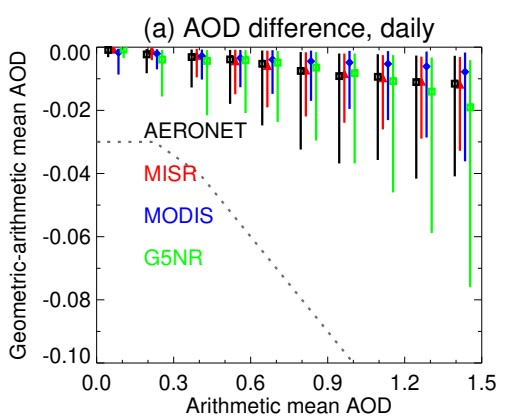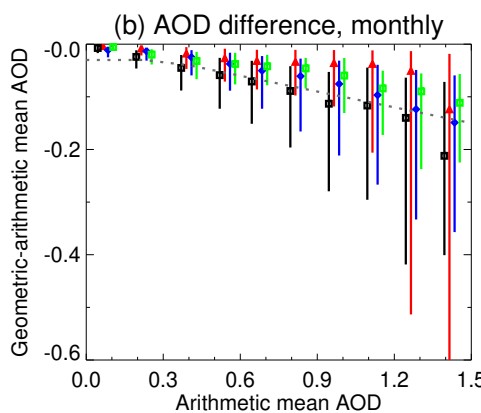

**Figure 8.** Median (symbols) and central 68 % (lines) of binned difference between geometric and arithmetic mean AOD ($\bar{\tau}_l - \bar{\tau}_n$) on daily (left) and monthly (right) time scales. Colours indicate AERONET (black), MISR (red), MODIS (blue), or G5NR (green) data. The AOD bin size is 0.15; data sets are horizontally offset slightly for clarity. The dashed grey lines indicate the GCOS goal AOD uncertainty of the maximum of 0.03 or 10 %.

20    The previous portion of the analysis has focused mostly on the occurrence and distinguishability of Normal and Lognormal

distributions for AOD; also relevant is the magnitudes of the differences introduced into the data sets by the choice of averaging

method and summary statistic. Figure 8 shows the difference between geometric and arithmetic mean AOD ($\bar{\tau}_n - \bar{\tau}_l$) binned

as a function of arithmetic mean AOD, on daily and monthly time scales, from the four data sets. For the daily plots the G5NR

temporal aggregation is shown, although results are similar for the spatial aggregation. Also shown on both panels is the Global

Climate Observing System (GCOS) goal uncertainty in AOD for an aerosol climate data record (CDR), of the greater of 0.03

or 10 % of the AOD (GCOS, 2011). AOD differences approaching or exceeding this level imply that aggregation method alone

causes a sensitivity of similar magnitude to the total desired uncertainty, and therefore that if data are to be aggregated then

choice of an appropriate technique is crucial.

It is important to realise that AOD difference $\bar{\tau}_l - \bar{\tau}_n$ is always zero or negative, as geometric means are always smaller

than or equal to arithmetic means (Cauchy, 1821). This means that the offsets will always be systematic. For daily data (left



panel of Figure 8), the median offset, and its dependence on AOD, are reasonably consistent between all four data sets. The

central 68 % of the observed offsets are also somewhat smaller than the GCOS uncertainty requirement, and below a total AOD
around 0.6, generally smaller than 0.02. Even a small change in reported AOD, if systematic, can have important implications
for calculations of climate forcing. This is particularly true for aerosol-cloud interactions, as these are very sensitive to both
the anthropogenic perturbation and the natural background state assumed. For example, using perturbed parameter simulations
to global climate models Carslaw et al. (2013) estimated that 45 % of the uncertainty in the global mean forcing due to the

cloud albedo effect of aerosols was related to uncertainties in the natural background aerosol burden, compared to 34 % for
anthropogenic emissions. Others, including Penner et al. (2011) and Grandey and Wang (2019), have similarly found large
dependence of forcing dependent on choice of background. Where AOD is low, such as over much of the global ocean, a small
absolute AOD change can be a large relative perturbation. Although the limitations of satellite retrievals for some of these
applications are well-known (e.g. Penner et al., 2011; Stier, 2016), the same argument may apply if forcing parametrisations

are developed from model simulations aggregated in certain ways. As a result, even differences smaller than the GCOS goal
uncertainty, such as the daily differences in Figure 8, may be significant for these purposes.

In contrast to the daily results, and even in low-moderate AOD loadings around 0.3, for monthly aggregates (right panel
of Figure 8) the difference is often similar to or larger than this GCOS uncertainty. This means that the choice of arithmetic
or geometric mean AOD as a summary metric in itself can and often does introduce systematic offsets in reported monthly

AOD of similar size to the goal total uncertainty for an AOD CDR. As with the daily data, the magnitude of the offset is
AOD-dependent; the magnitude is, however, less consistent than for the daily results, with the median offset being largest for
AERONET. This might in part reflect known tendencies for a high bias in low-AOD conditions and/or low bias in high-AOD
conditions in these satellite products (Kahn et al., 2010; Levy et al., 2013; Sayer et al., 2019), meaning that the difference
$\bar{\tau}_l - \bar{\tau}_n$ is dampened due to diminished spatial and temporal variability within and between days.

5 The asymmetry of the variabilities (vertical lines) in both panels of Figure 8 indicates a dependence of the difference on
the specific local conditions. This implies that no simple scaling correction can be applied to existing data sets to transform
between arithmetic and geometric estimates and instead the analyst should recompute from the source data.

As Figure 8 established that on monthly time scales sensitivities to averaging method often exceed GCOS goal CDR uncer-
tainties, Figure 9 maps how frequently such exceedences occur. The behaviour for seasonal aggregates (not shown) is more

10 pronounced than that of monthly, and shows similar spatial features. As seen in earlier parts of this study, the four data sets give
broadly consistent spatial patterns, but differences in magnitude. Specifically, this is seen most frequently (30-90 %, dependent
on grid cell and data set) in eastern Asia and Saharan outflow regions, which is unfortunate because these are important and
frequently-studied components of the global aerosol system. Exceedence of GCOS thresholds in 10-40 % of months is also
seen fairly consistently across much of eastern North America and Eurasia, South America, South-eastern Asia, and Southern

15 Africa. This is most common during the summer months (former two cases) and local biomass burning seasons (other cases)
when AOD levels are generally higher. GCOS threshold exceedence is infrequent (observed <10 % of the time) over the re-
mote open ocean in any of the data sets, although may be slightly elevated for oceanic regions downwind of continental aerosol
sources. In all of these regions, the monthly data show higher consistency with Lognormality more often than they do than

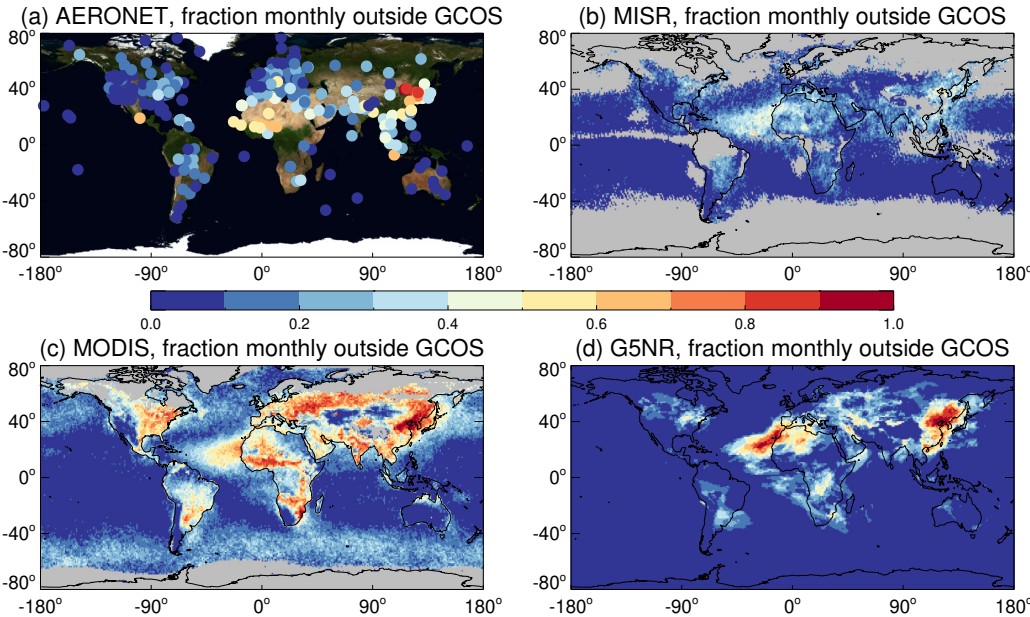

**Figure 9.** Fraction of months where the difference between arithmetic and geometric mean AOD is larger than the GCOS goal uncertainty for an AOD climate data record, i.e. $|\bar{\tau}_l - \bar{\tau}_n| \geq \max[0.03, 0.1\bar{\tau}_l]$. Panels show results for (a) AERONET, (b) MISR, (c) MODIS, and (d) G5NR.

Normality (Figure 6), particularly for the AERONET record, which has the most reliable AOD. Therefore, the locations where the difference between $\bar{\tau}_n$ and $\bar{\tau}_l$ is largest are also generally those where the data support Lognormal summary statistics the most.

## 4.2 Implications for AOD trend analyses

As the differences between arithmetic and geometric mean are larger for higher-AOD regions (Figures 8, 9) choice of summary statistic could also influence the calculation of AOD trends. Specifically, as $\bar{\tau}_l < \bar{\tau}_n$ by an increasing amount as AOD increases, smaller magnitudes of calculated trends would be expected (as the maxima are dampened to a higher degree than the minima). Multiple studies over the past few decades have looked at AOD trends globally and regionally, whether over oceans only (e.g. Mishchenko and Geogdzhayev, 2007; Zhao et al., 2008; Thomas et al., 2010; Zhang and Reid, 2010; Li et al., 2014a) or both oceans and land (e.g. Hsu et al., 2012; Chin et al., 2014; Li et al., 2014b; Yoon et al., 2014; Pozzer et al., 2015; Klingmüller et al., 2016). While data sources, periods of analysis, and analysis techniques differ, as do quantitative results, several features tend to be consistently reported:

1. AOD over the global ocean, and over many ocean basins, has not changed very much;

2. AOD over parts of eastern North America and Europe has decreased in recent decades; and





**Table 3.** Decadal trends ($\pm 1\sigma$ uncertainty estimates) in AERONET AOD at 550 nm, estimated using arithmetic and geometric mean AOD as a basis for time series.

| Trend type | Record length* | From mean AOD Trend, AOD dec$^{-1}$ | $\chi^2$ | From geometric mean AOD Trend, AOD dec$^{-1}$ | $\chi^2$ |
|---|---|---|---|---|---|
| \multicolumn{6}{c}{Ascension Island (7.97639° S, 14.4147° W)} |
| Monthly | 153 | -0.001 ($\pm$0.005) | 468 | 0.000 ($\pm$0.005) | 447 |
| DJF | 16 | -0.007 ($\pm$0.006) | 52.0 | -0.002 ($\pm$0.007) | 60.8 |
| MAM | 15 | -0.008 ($\pm$0.006) | 64.6 | -0.008 ($\pm$0.006) | 55.0 |
| JJA | 14 | 0.004 ($\pm$0.010) | 57.8 | 0.006($\pm$0.010) | 39.2 |
| SON | 14 | 0.042 ($\pm$0.007) | 89.2 | 0.032 ($\pm$0.007) | 83.2 |
| \multicolumn{6}{c}{GSFC (38.9925° N, 76.8398° W)} |
| Monthly | 287 | -0.018 ($\pm$0.003) | 931 | -0.017 ($\pm$0.003) | 734 |
| DJF | 24 | -0.013 ($\pm$0.006) | 18.4 | -0.009 ($\pm$0.005) | 11.2 |
| MAM | 26 | -0.039 ($\pm$0.008) | 109 | -0.031 ($\pm$0.006) | 80.6 |
| JJA | 25 | -0.144 ($\pm$0.021) | 79.8 | -0.095 ($\pm$0.013) | 67.6 |
| SON | 24 | -0.021 ($\pm$0.005) | 55.4 | -0.015 ($\pm$0.002) | 48.9 |
| \multicolumn{6}{c}{Solar Village (24.9069° N, 46.3973° E)} |
| Monthly | 158 | 0.093 ($\pm$0.024) | 476 | 0.080 ($\pm$0.022) | 505 |
| DJF | 13 | 0.104 ($\pm$0.042) | 48.6 | 0.076 ($\pm$0.032) | 60.1 |
| MAM | 15 | 0.218 ($\pm$0.040) | 102 | 0.176 ($\pm$0.034) | 108 |
| JJA | 14 | 0.176 ($\pm$0.043) | 130 | 0.156 ($\pm$0.042) | 136 |
| SON | 14 | 0.048 ($\pm$0.019) | 92.6 | 0.041 ($\pm$0.019) | 114 |

*Number of contributing months for the monthly time series; number of contributing years for the seasonal time series

3. Among the strongest positive AOD changes tend to be seen over the Arabian Peninsula.

Using three long-term AERONET sites (one for each of the above features), Table 3 provides decadal AOD trends calculated using geometric mean AOD $\bar{\tau}_l$ and arithmetic mean AOD $\bar{\tau}_n$ as a basis. These sites were used in some of the above studies to complement satellite retrieval/model simulation analyses; in all cases, those studies used arithmetic mean AOD. Ascension Island is in the south Atlantic Ocean where reported AOD trends are typically small, and presently has data available from 1998-2016. Goddard Space Flight Center (GSFC) in Maryland, USA, a region of decreasing AOD, has data from 1993 onwards and is one of the longest-running AERONET sites; Solar Village (operated 1999-2013) was at a solar power farm northwest of Riyadh, Saudi Arabia, and near the maximum of AOD trends reported in previous studies. The purpose here is not to perform an exhaustive global trend analysis, but to assess quantitatively the implications of Lognormally-distributed AOD on some well-reported features of global aerosol trends.





These studies typically calculated trends based on deseasonalised monthly mean AOD time series, and calculating a linear

least-squares regression fit. Deaseasonalisation was achieved either by subtracting the mean AOD annual cycle over the time

period, or (Thomas et al., 2010; Klingmüller et al., 2016) by using harmonic regression to model the annual cycle. Li et al.

(2014a, b) took a somewhat different approach by analysing temporal variability of principal components of monthly AOD

fields, rather than the AOD fields themselves. In some of these analyses, seasonal trends were calculated by averaging the

monthly data within each season, although in this case it is arguably more reasonable to use seasonal aggregates from daily data

as a basis. The motivation for considering seasonal trends is that some aerosol features, and their variability, are characteristic

of particular seasons (e.g. Ascension Island and GSFC sample transported smoke through summer and autumn but seldom

during other seasons; at Solar Village dust storms are most frequent and intense in spring and summer).

Here, linear trends are calculated using both monthly and seasonal aggregates, for both Normal (i.e. arithmetic mean) and

Lognormal (i.e. geometric mean) aggregates, both calculated from daily AOD ($\bar{\tau}_n$ or $\bar{\tau}_l$ respectively). Monthly trends are

calculated using the monthly AOD time series, after subtraction of the mean seasonal cycle, as in previous studies. Seasonal

trends do not require a deseasonalisation step. The data are fit using linear least-squares regression, with the weights equal

to the standard error on the estimated monthly (or seasonal) AOD. For the Lognormal averaging this is strictly asymmetric,

although it is approximated as symmetric in this case, which has negligible influence on the results. The lower limit for these

standard errors is taken as 0.01, corresponding to the AERONET AOD uncertainty. As this is largely dominated by calibration

uncertainty (Eck et al., 1999) it is not significantly reduced by averaging and can therefore be considered systematic over a

single (roughly year-long) deployment, but closer to random over a multi-year time series. The standard error on the annual

cycle is added in quadrature to the estimated uncertainty on the monthly time series, to account for the uncertainty in the

deseasonalisation step. Following Weatherhead et al. (1998), the lag-1 autocorrelation is estimated and used to adjust the

uncertainty estimates. Further, the $\chi^2$ statistics on the fits, which have an expected value of $n$-2 (where $n$ is the record length

and two parameters are fit in the regression), were in most cases somewhat in excess of this (Table 3). This implies that the

standard errors are not a complete representation of the uncertainty on the time series data, and/or that a linear model does not

fully describe the variation. Thus, in addition to the autocorrelation correction, the trend uncertainty estimates in Table 3 are

also scaled by $\sqrt{(\chi^2/(n-2))}$.

At each of the three sites, the decadal AOD trends are qualitatively the same whether calculated using arithmetic or geometric

mean AOD time series as a basis. However, as expected, trends using geometric mean AOD are smaller in magnitude (i.e.

increases and declines in AOD are less pronounced). The decrease in magnitude is often of order 10-30 %, which is typically

within the $1\sigma$ uncertainty estimates on the calculated AOD trend. The implication of this is that, as a result of assuming an

underlying Normal distribution, prior studies may be qualitatively correct on the sign of AOD trends, but quantitatively have a

tendency to overestimate their magnitude.

## 4.3    Summary and recommendations for data use

Widely-used spatiotemporal aggregates of aerosol data from surface observations, satellite retrievals, and model simulations

typically consist of arithmetic means and standard deviations of finer-resolution data. These statistics are most meaningful





for Normally-distributed data, while previous work has indicated that AOD is often distributed close to Lognormally on large
scales. This study has illustrated the use of Shapiro-Wilk tests as a comparative tool to assess whether quantities such as AOD
are more consistent with draws from Normally or Lognormally distributed populations. Data from ground-based observations
(AERONET), satellite retrievals (MISR, MODIS) and model simulations (G5NR) provide broadly consistent results. As time
scales increase from days to months to seasons, data become increasingly more consistent with Lognormal than Normal distri-

butions, and the differences between arithmetic and geometric mean AOD become larger; assuming Normality systematically
overstates both the typical level of AOD and its variability. In low-AOD regions such as the open ocean and mountains, often
the AOD difference is sufficiently small (<0.01) as to be unimportant for many applications, especially on daily timescales.
However, in continental outflow regions and near source regions over land, and on monthly or seasonal time scales, the dif-
ference is frequently larger than the GCOS goal uncertainty on a climate data record (the larger of 0.03 or 10 %). As a result

of this, estimated trends in geometric mean AOD are smaller in magnitude than (although consistent in sign with) those in
arithmetic mean AOD.

The main recommendations from this study for future missions and reprocessing of current data sets/simulations are as
follows:

1. The frequency distribution of a geophysical quantity should be analysed in order to asses how best to aggregate it. This

25        analysis should be done at the spatial and temporal scale(s) of interest for the aggregation, because distributions are
scale-dependent. The Shapiro-Wilk technique is a powerful tool to assess discrepancies from a Normal or Lognormal
distribution, and should be further combined with desired performance thresholds to assess whether discrepancies are
scientifically relevant for a given quantity.

2. Ideally AOD aggregates such as satellite L3 products, but also from ground-based (e.g. AERONET) and model simu-

30        lations, should report geometric mean or median rather than (or in addition to) arithmetic mean AOD. This is because
multiple data records provide evidence that AOD distributions are generally closer to Lognormal than Normal, partic-
ularly on monthly and seasonal time scales, and geometric mean is the more natural and meaningful summary statistic
for such data. Geometric mean AOD is systematically lower, often (on monthly/seasonal time scales) by more than
the GCOS goal climate data record uncertainty of the larger of 0.03 or 10 %, so the choice of averaging method is
scientifically important.

3. Due to the computational burden required on the data producer or user's end (i.e. for satellites, obtaining the full L2 data
record to reaggregate to daily and then monthly time steps), this is unlikely to happen in the short term. In the meantime,

5        calculation of geometric mean monthly aggregates from current standard (i.e. arithmetic mean) daily L3 aggregates
could be a useful stopgap measure. This is because the volume of daily L3 data is smaller than L2, and daily spatial
aggregates were found to be less sensitive than monthly to choice of arithmetic vs. geometric averaging.

4. Comparisons and statistical assessments of AOD must account for the expected numerical distribution. Some com-
mon performance assessment techniques making use of sum-of-squared calculations, such as root mean square error





or coefficient of determination, should not be used in all cases as they can be systematically skewed by large tails on non-Normally distributed data (Seegers et al., 2018).

The analysis presented here refers to AOD, but the methodology is general. GCOS (2011) provide goal uncertainties for many geophysical CDRs, which may be helpful for assessing the importance of averaging method in different disciplines. Overall the Lognormal distribution seems a better reference for AOD aggregates than the Normal distribution, on spatial scales
of single locations or 1 degree, and temporal scales from days to seasons.

It is important to bear in mind that these simple distribution forms are just approximations for the true underlying distribution of a geophysical quantity, and the relevant problem is in identifying one which is a sufficiently accurate representation for a given task. Normal and Lognormal distributions are mathematically convenient and represent many data sets reasonably well, which is a motivating factor for considering these two both historically and in the present work. Sometimes multiple
distribution forms are suitable: this analysis has shown that often in low-AOD conditions the choice of Normal or Lognormal representation may not matter for many purposes. Further, while not analysed here, dependent on choice of parameters Gamma distributions (often used to describe cloud particle size distributions, e.g. Platnick et al., 2017) can be numerically similar to Lognormal. Sometimes multiple modes are required, and sometimes neither distribution is a suitable approximation. If only a few distributions or points need to be summarised, then it is of course preferable to show the actual distributions and/or an
informative summary which is agnostic to any particular distribution shape, such as a box-whiskers plot. However for many larger-scale analyses, aggregated outputs from observations and model simulations are likely to remain the format of choice for many data users, due to their convenience and significantly lower computational/storage requirements than full-resolution (e.g. L2) data. The above recommendations will result in more statistically and scientifically meaningful data sets, and decrease potential systematic biases which can lead to erroneous qualitative and quantitative interpretation about the state of the Earth
system.

*Data availability.*  The geometric mean AOD output presented in this work are available upon request to the authors. Information on the availability of the input data sets used is provided in the Acknowledgements.

*Author contributions.*  AMS and KDK jointly conceptualised the analysis. AMS performed the analysis and drafted the paper. Both authors contributed to the editing of the text.

*Competing interests.*  The authors declare no competing interests.



*Acknowledgements.* This research was performed as part of NASA Plankton, Aerosol, Cloud, ocean Ecosystem (PACE) mission develop-
5    ment. AERONET data are available from https://aeronet.gsfc.nasa.gov; the AERONET team and site Principal Investigators and managers are
thanked for the creation and maintenance of the AERONET data record. MISR and MODIS data are available from https://earthdata.nasa.gov,
and G5NR simulation output from https://g5nr.nccs.nasa.gov/data/. Satellite retrieval and modeling teams, and hosting entities, are acknowl-
edged for the development and archiving of these data sets. T. F. Eck (USRA), B. N. Holben (NASA GSFC), and A. Smirnov (SSAI) are
thanked for useful discussions about early AOD/turbidity measurement networks, and their strengths and limitations. P. Castellanos (NASA
10  GSFC) is thanked for advice on the use of the G5NR simulation. C. J. Merchant (University of Reading) is thanked for input on uncertainty
characterisation in sea surface temperature data.



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
