# Peer review of "How should we aggregate data? Methods accounting for the numerical distributions, with an assessment of aerosol optical depth"

_Atmospheric Chemistry and Physics, 2019_

## Short Comment (SC1) · 13 Aug 2019

I wish to support the publication of this manuscript. As the authors outline, it has long been known that aerosol optical depth is not normally distributed, such that an arithmetic mean is not expected to represent real-world behaviour. This paper will hopefully remind the community of the implications of that fact and encourage greater use of geometric means in analysis and logarithmic scales in figures.

If I may comment on Figs. 2 and 3, it does not seem surprising that the majority of

the planet exhibits little difference between the daily arithmetic and geometric means. The plot below shows the difference between the geometric and arithmetic means if we generate random, lognormally distirbuted data for a range of medians and widths. Your threshold of -0.01 is not exceeded for distributions with a range of small widths and medians that are common in nature.

This begs the question why the difference does matter in Fig. 5. When observing complex aerosol environments, such as the Saharan outflow, the satellite likely samples a single population of aerosol on any given day, which is lognormally distributed. Over a month, several populations are sampled, giving a multimodal distribution. Geometric statistics are more appropriate for combining these samples and so the lognormal distribution is found to be superior. Conversely, over Australia, where MODIS retrieves a very narrow range of AODs, the difference is still found to be negligible.

Alternatively, the increased data volume highlights the failings of arithmetic statistics because too few very low AODs and too many very high AODs are observed for a normal distribution. When there are fewer observations, it is harder for the Shapiro-Wilk test to discriminate behaviour in the distribution's wings.

In summary, I wonder if a single lognormal distribution may not sufficient in many circumstances or if the problem is more that AOD must be postive, rather than an intrisic lognormality? I don't believe these details affect the authors central point that geometric statistics should be used to evaluate AOD but am curious of their opinion.

I also include some technical comments and corrections. P1L2 means line 2 of page 1.

P2L17  in some cases they have also been

P2L21  a regular grid and so are often more

P2L24  observe every location at all the times.

P2L26  and sometimes is not negligible

P7L10  example application is to AOD

P19L20  are most common in so-called

P22L21  also relevant are the magnitude of the differences

P30L29  The page number is 2.

P31L14  The DOI is 10.1029/1999JD900923.

 P32L2  The page numbers are 2276-2295.

P32L18  The page numbers are 4026-4053.

P34L13  The page numbers are 13,404-13,408.

P34L21  The page numbers are 672-676.

P35L14  The page numbers are 13,965-13,989.

P36L21  The page numbers are 429-439.

The following Python code was used to generate the figure above,

```
import matplotlib.pyplot as plt
import numpy as np
from scipy.stats import norm, lognorm
from itertools import product

log_delta = []
for mean, unc in product(np.arange(0.01, 1.0, 0.01), np.arange(0.01, 0.5, 0.01)):
```

```
    sample = lognorm.rvs(unc, scale=mean, size=10000)
    geometric_mean = np.exp(np.mean(np.log(sample)))
    log_delta.append(geometric_mean - sample.mean())

log_delta = np.array(log_delta).reshape((99, 49))

yy, xx = np.meshgrid(np.arange(0.005, 0.5, 0.01), np.arange(0.005, 1.0, 0.01))

ax = plt.axes()
im = ax.pcolormesh(xx, yy, log_delta, vmin=-0.02, vmax=0, cmap="coolwarm")
ax.set_ylabel("Geometric width")
ax.set_xlabel("Distribution median")
ax.set_title("Lognormal distribution")
plt.colorbar(im, label="Geometric - arithmetic mean")

plt.show()
```

Lognormal distribution

[Figure]

**Fig. 1.**

---

## Short Comment (SC2) · 13 Aug 2019

I'd like to thank Dr. Povey for these thoughtful comments - this sort of thing is exactly the type of discussion we were hoping to provoke.

I agree with Dr. Povey's thinking that we need to have some deeper discussion of what we intend users to take away when we provide them with aggregates and how to best provide concise, useful information while decreasing the chance of misunderstanding or misuse. My personal opinion is that the geometric mean (lognormal statistics) is a

better summary statistic than arithmetic mean (normal statistics) for AOD aggregates, overall, for those cases when there is likely only a single mode (due to the skew and nonnegativity). In the case of multimodal distributions, the answer is less clear. Perhaps it lies in decreasing aggregation scales so that these become less frequent (e.g. encourage use of daily rather than longer-term composites). The idea of providing fit statistics for multiple modes as outlined in Povey & Grainger (2019) is also a good one, although requires extra thought on the user end, as would providing histograms directly (alongside a data volume issue). I look forward to carry these discussions on with the broader community through AeroCom and AeroSat.

The Figure included in the comment is an interesting one (and essentially a multidimensional version of our Figure 1). I will consider creating something like it for the revised manuscript, as it could be quite informative to overplot typical values for certain AERONET sites on the top. I will note that since Dr. Povey's example is calculated in natural log while our example was in base 10 log, the geometric standard deviations covered in Dr. Povey's simulations should be decreased by a factor of ln 10 (about 2.3) to be equivalent, i.e. our geometric standard deviation of 0.35 would correspond to about 0.8 (off the top of the scale) in his plot.

Thanks also for the grammatical/reference comments, which I'll check and correct as necessary.

---

## Referee Comment (RC1) · Anonymous Referee #3 · 10 Sep 2019

This paper looks at an important yet widely neglected issue in selecting the appropriate summary statistics to create daily or monthly climatology from instantaneous measurements. AOD is used as an example, but the study is applicable to any geophysical variables. The study is well thought out, carefully executed, and clearly presented. I have a few comments below. 1. It would be appealing to give plausible explanations, from the standpoint of physical mechanisms, to the fact that certain geophysical variables, like AOD, follow lognormal distribution (raindrop size is often described by Gamma distribution, which has a similar skewed shape to lognormal), while some are

Gaussian. I find the authors' attempt at P6L5 unconvincing. Statistically, lognormal distribution arises from multiplicative processes while normal distribution is from summation of independent/identically distributed (Gaussian or non-Gaussian) processes (central limit theorem). But I feel that it is hard to relate this statistical interpretation to the physical processes happening in reality. The reason for the skewed distribution of many geophysical variables may be due to the simple fact that they are positively defined. This is also supported by the authors' analysis that in clean conditions (AOD approaching positively zero), geometric and arithmetic means are not all that different. 2. Negative or zero AODs are set to a very small positive value. This has to be done in order to calculate geometric mean. I wonder in such special cases (even if they are rare), does it incur any arbitrary bias that renders use of geometric mean less meaningful compared to arithmetic mean? How would the operational L2-to-L3 algorithm deal with the negative or zero retrievals? 3. A possible development for aerosol product is to collate products from different sensors for the overlapping domain (such as the MODIS twins from Terra and Aqua). In that case, which mean is more appropriate, geometric or arithmetic? 4. The study discussed the impact of geometric vs arithmetic means on trend analysis. How about the aerosol climatic impact? Are L3 products used in climate model evaluation or assimilation that the choice of aggregation method may have important effects?
* * *

---

## Short Comment (SC4) · 11 Sep 2019

I enjoyed reading the manuscript. This study presents interesting results on AOD distributions in grid, and how to reprocess the data. I was very surprised that bimodal distribution (Figure 5) can also be fit as Lognormality. This method is really useful for me! We published a paper about cloud optical depth distribution patterns recently. Maybe the authors want to consider referencing: Chen YL, Chong KZ, Fu YF. 2019. Impacts of distribution patterns of cloud optical depth on the calculation of radiative forcing. Atmos. Res. 218: 70-77. doi: 10.1016/j.atmosres.2018.11.007

---

## Referee Comment (RC2) · Anonymous Referee #1 · 17 Sep 2019

The authors discuss the nature of probability distributions of AOD, the aerosol optical depth. The authors start by analysing for a diverse collection of datasets (AERONET, MODIS, MISR, GEOS5 Nature Run), whether spatially or temporally grouped data is better described by a normal or log-normal distribution. They show that at short time-scales (day), the normal distribution is appropriate but that at longer time-scales, log-normal distributions are more realistic. They then continue to show that means derived from such datasets, using either arithmic or geomtric means, can be quite different. In particular, they show that trend estimate can differ significantly in magnitude (though

not in sign). As the data mostly exhibit log-normal distributions at longer time-scales, the authors conclude that the common use of arithmic means in trend analysis is inappropriate. The paper concludes with suggestions for improvements in aggregation methods. This is an interesting paper about a fundamental issue in Earth sciences and entirely appropriate to AMT.

The statistical analysis in this paper seems sound. However, I feel the authors may be overstating the importance of this issue. Often we don't look at AOD but at differences in AOD (satellite evaluation, model evaluation, changes between present day and pre-industrial, model sensitivity studies, etc). The resulting distributions are in my experience usually more normally than log-normally distributed.

Also, any log-normal distribution can be accurately defined by arithmic mean and standard deviation. Actually, there is a one-to-one transformation from the arithmic statistics to the geometric statistics, see e.g. https://en.wikipedia.org/wiki/Lognormal_distribution. My interpretation is that it is not important whether one uses arithmic or geometric statistics, as long as one is aware that their use does not imply (!) either a normal or log-normal distribution.

Another issue is physical conservation of the property under study. AOD is not a good example so let's consider column burdens of aerosol or trace gases. These may be expected to have log-normal distributions in time and space as well. Describing them with geometric means would cause loss of mass conservation! Consider a dataset at 10 km that is aggregated to 100 km: the arithmic mean preserves total mass in the 100 km grid-box while the geometric mean does not.

It seems one has to consider what is causing the log-normality: if it is due to log-normal retrieval errors, geometric means seem justifiable as they ameliorate the effect of outliers. If it is due to the nature of the property, conservation-laws may be more important and arithmic means are to be used. I am sure much more can be said about this.

That said, if arithmic means are used to describe log-normal distributions and then carried forward through non-linear analyses under the assumption of normality, significant problems may arise. The authors allude to this on p. 23, l 19 when they talk about parametrisations.

It would be great if the authors take the above into consideration when preparing their final manuscript. In all this is a worthwhile discussion.

Minor comments:

p 3, l 33: it would be good to state the relation between arithmic mean (and stddev) with geometric mean (and stddev) for a log-normal distribution. Such relation exists, see https://en.wikipedia.org/wiki/Log-normal_distribution

p 5, l 17: "will overstate the typical level of AOD observed and its variability" . While I understand the authors' intention, it seems to me this sentence suffers from the absence of what is "typical". It would appear that "typical" here refers to the geometric mean as a parameter that defines a log-normal distribution. However, there is a simple 1-on-1 mathematical relation between arithmic and geometric mean of a log-normal distribution. I.e. the arithmic mean defines a log-normal distribution as well as the geometric mean. Hence both arithmic and geometric mean can be used to define what is "typical".

p 6, l 25: "using Normal-appropriate statistics has systematic quantitative implications for the interpretation of the data." Only if the arithmic mean and stddev are interpreted as defining a normal distribution. It is perfectly possible to calculate both without reference to a normal distribution. Actually, they can define a log-normal as well.

p 6, l3: "these factors may include": I believe turbulence is an important factor in the creation of log-normal distributions?

p 8, l 13: "This quadratic formulation is more robust to calibration problems in individual channels" more robust than what? Maybe consider dropping "more"?

p 11, l 7: "tail-waited" tail-weighted ?

p 12, l 3: Shouldn't arithmic and geometric stddev be compared as well?

p 13, l 3: "results for temporal (from AERONET and G5NR) and spatial (from MISR, MODIS, and G5NR) frequency distributions of" This confused me as both Figures show spatial distributions of the WS test. The test, in all cases, was presumably done on time-series of data. The captions to the figures seem to say something different: either data was a temporal aggregate (which suggests G5NR results are at its native resolution) or spatially aggregate (which suggests each 30 min of G5NR data was used). Please clarify this?

p 14, l 5: "calculating an arithmetic mean when the underlying distribution is Lognormal (or vice-versa) introduces an error smaller than 0.01." I disagree with the use of the word 'error'. No error is incurred at all. It is always possible to calculate arithmic means. Any error is due to limited sample size. See also my previous comments.

p 17, l 7: "Note also that the near-universal choice of aggregating daily on a UTC calendar day basis, rather in terms of local solar time, can further complicate matters for locations far from the meridian." For another example, see Schutgens, Partridge & Stier ACP 2016, Fig. 13 & 14.

p 23 , l 11: "Even a small change in reported AOD, if systematic, can have important implications for calculations of climate forcing." But changes (differences) in AOD are far more likely to have a normal distribution.

p 23, l 20: "the same argument may apply if forcing parametrisations are developed from model simulations aggregated in certain ways" This is a fair point. A lot of studies point out the distorting impact of non-linear physics/chemistry when using just the mean to represent a distribution. One example from remote sensing is the plane-parallel bias noted in cloud retrievals of LWP. Note however that such biases exist not because of an arithmetic mean but the representation of any disribution by a single number.

p 23, l 6: "This implies that no simple scaling correction can be applied to existing data sets to transform between arithmetic and geometric estimates" . Assuming a normal or log-normal distribution, exact transformations exist between mean and stddev of arithmic and geometric statistics.

Sect 4.2 The analysis in this section seems sound and I have no problems with it. That arithmic means yield different trends than geometric means is no surprise, after all these are different means. However, there is the suggestion that geometric means are better simply because the underlying distribution is log-normal. Rather, geometric statistics make it easier to interpret changes in a log-normal distribution but they do not provide more information (or put differently: the arithmetic statistics are not "wrong"). Note also that trend analysis of changing log-normal distribution really requires geometric stddev to be analysed as well but this is seldom done.

p 26, l 6: "but quantitatively have a tendency to overestimate their magnitude." It may be good to repeat here that at these three sites the log-normal distribution is the more appropriate distribution to use (previous analysis, Sect 3). At least that seems to be the suggestion here?

p 27, l 20: "estimated trends in geometric mean AOD are smaller in magnitude" Trends in satellite data are over often calculated over regions, not like the point sources the authors have used in their example. I wonder how this will affect these conclusions? At some point the central-limit-theorem should kick in and turn any log-normal distribution into a normal one?

p 27, point 2: this point seems to imply it is ok to average AOD in time and compare satellites with satellites or models, as long as we use the proper mean. The authors know that different sampling of data sources often has a far bigger impact. Maybe it is good to state that here.

p 27, l 9: "root mean square error" This is a difference between two AOD and is likely to be normally distributed.

---

## Author Comment (AC1) · 1 Oct 2019

We would like to think Drs. Chen, Povey, and two anonymous reviewers for their suggestions on our manuscript. We appreciate the level of detail and nuance in the comments. We have prepared a revised version with attention to these comments. In the below, text in **bold** indicates Short Comments or Referee Comments, and that in regular type indicates our response.

In addition to responses to these comments, the following other changes have been made to the manuscript:
- Corrected a typographical error in Equations 1 and 2 where the normalisation factor N appeared on both the left and right-hand sides of the equations.
- Rearranged contents between the Data Availability and Acknowledgments sections to conform with Copernicus style requirements.

**Short comment from Adam Povey**

**I wish to support the publication of this manuscript. As the authors outline, it has long been known that aerosol optical depth is not normally distributed, such that an arithmetic mean is not expected to represent real-world behaviour. This paper will hopefully remind the community of the implications of that fact and encourage greater use of geometric means in analysis and logarithmic scales in figures.**

We thank Dr. Povey for his support of our study.

**If I may comment on Figs. 2 and 3, it does not seem surprising that the majority of the planet exhibits little difference between the daily arithmetic and geometric means. The plot below shows the difference between the geometric and arithmetic means if we generate random, lognormally distirbuted data for a range of medians and widths. Your threshold of -0.01 is not exceeded for distributions with a range of small widths and medians that are common in nature. This begs the question why the difference does matter in Fig. 5. When observing complex aerosol environments, such as the Saharan outflow, the satellite likely samples a single population of aerosol on any given day, which is lognormally distributed. Over a month, several populations are sampled, giving a multimodal distribution. Geometric statistics are more appropriate for combining these samples and so the lognormal distribution is found to be superior. Conversely, over Australia, where MODIS retrieves a very narrow range of AODs, the difference is still found to be negligible. Alternatively, the increased data volume highlights the failings of arithmetic statistics because too few very low AODs and too many very high AODs are observed for a normal distribution. When there are fewer observations, it is harder for the Shapiro-Wilk test to discriminate behaviour in the distribution's wings. In summary, I wonder if a single lognormal distribution may not sufficient in many circumstances or if the problem is more that AOD must be postive, rather than an intrisic lognormality? I don't believe these details affect the authors central point that geometric statistics should be used to evaluate AOD but am curious of their opinion.**

As we posted in the Discussion forum these thoughtful comments are exactly the type of discussion we were hoping to provoke. We agree with Dr. Povey's thinking that we need to have some deeper discussion of what we intend users to take away when we provide them with aggregates, and how to best provide concise, useful information while decreasing the chance of misunderstanding or misuse. Realistically most users want and expect only a single number to use in their analysis and are not always equipped to consider that this is a summary of a distribution of values. My personal opinion is that the geometric mean (lognormal statistics) is a better summary statistic than arithmetic mean (normal statistics) for AOD aggregates, overall, for those cases when there is likely only a single mode (due to the skew and nonnegativity). In the case of multimodal distributions, the answer is less clear. Perhaps it lies in decreasing aggregation scales so that these become less frequent (e.g. encourage use of daily rather than longer-term composites). The idea of providing fit statistics for multiple modes as outlined in Povey & Grainger (2019) is also a good one, although requires extra thought on the user end, as would providing histograms directly (which would also create a data volume issue). So this is in an open question, and we (as well as Dr. Povey in the aforementioned paper) have provided steps towards possible solutions.

The Figure included in the comment is an interesting one (and essentially a multidimensional version of our Figure 1). We have added a similar Figure in our revised manuscript (now Figure 2). Note that since Dr. Povey's example is calculated in natural log while our example was in base 10 log, the geometric standard deviations covered in Dr. Povey's simulations should be decreased by a factor of ln 10 (about 2.3) to be equivalent, i.e. our geometric standard deviation of 0.35 would correspond to about 0.8 (off the top of the scale) in his plot. This may

resolve the comment comparing Figures 2, 3, and 5. Choice of logarithmic base (and conversions between them) is further discussed in the text between Equations 2 and 4.

**I also include some technical comments and corrections. P1L2 means line 2 of page 1.**

**P2L17 in some cases they have also been**
**P2L21 a regular grid and so are often more**
**P2L24 observe every location at all the times.**
**P2L26 and sometimes is not negligible**
**P7L10 example application is to AOD**
**P19L20 are most common in so-called**
**P22L21 also relevant are the magnitude of the differences**
**P30L29 The page number is 2.**
**P31L14 The DOI is 10.1029/1999JD900923.**
**P32L2 The page numbers are 2276-2295.**
**P32L18 The page numbers are 4026-4053.**
**P34L13 The page numbers are 13,404-13,408.**
**P34L21 The page numbers are 672-676.**
**P35L14 The page numbers are 13,965-13,989.**
**P36L21 The page numbers are 429-439**

We have checked and corrected the above.

**Short comment from Yilun Chen**

**I enjoyed reading the manuscript. This study presents interesting results on AOD distributions in grid, and how to reprocess the data. I was very surprised that bimodal distribution (Figure 5) can also be fit as Lognormality. This method is really useful for me! We published a paper about cloud optical depth distribution patterns recently. Maybe the authors want to consider referencing: Chen YL, Chong KZ, Fu YF. 2019. Impacts of distribution patterns of cloud optical depth on the calculation of radiative forcing. Atmos. Res. 218: 70-77. doi:10.1016/j.atmosres.2018.11.007**

Thank you for the kind words, and the reference. That paper ties into one of the other reviewer comments about climate impacts, by showing that distribution shape affects inferred forcing (at least for clouds). For aerosols the numerical values will be different, but a similar principle is likely to hold. We've cited this paper in the expanded discussion in the revised manuscript.

**Review by Anonymous Referee #1**

**The authors discuss the nature of probability distributions of AOD, the aerosol optical depth. The authors start by analysing for a diverse collection of datasets (AERONET, MODIS, MISR, GEOS5 Nature Run), whether spatially or temporally grouped data is better described by a normal or log-normal distribution. They show that at short timescales (day), the normal distribution is appropriate but that at longer time-scales, lognormal distributions are more realistic. They then continue to show that means derived from such datasets, using either arithmic or geomtric means, can be quite different. In particular, they show that trend estimate can differ significantly in magnitude (though not in sign). As the data mostly exhibit log-normal distributions at longer time-scales, the authors conclude that the common use of arithmic means in trend analysis is inappropriate. The paper concludes with suggestions for improvements in aggregation methods. This is an interesting paper about a fundamental issue in Earth sciences and entirely appropriate to AMT.**

We are pleased that the referee finds the study interesting and appropriate to the journal.

**The statistical analysis in this paper seems sound. However, I feel the authors may be overstating the importance of this issue. Often we don't look at AOD but at differences in AOD (satellite evaluation, model evaluation, changes between present day and pre-industrial, model sensitivity studies, etc). The resulting distributions are in my experience usually more normally than log-normally distributed.**

We partially agree with the reviewer here. Yes, it is likely that differences between estimates will be similar whether calculated as arithmetic or geometric mean (i.e. if you compare the differences in two monthly geometric mean AODs from some products, they are likely to be similar to the differences in monthly arithmetic means), and the distribution of differences is likely to be closer to Normal. However, (1) that is not a given, and (2) a large number of papers are not about comparing different AOD composites (for example many take AERONET or satellite data at a given location and you see statements like "the mean AOD was X+/-Y" without consideration of the shape of the distribution. It is in these latter cases where we feel reporting either geometric mean and standard deviation, or median and percentiles of the distribution, would be more meaningful. We have added a note in the expanded discussion sections of the paper about AOD differences.

**Also, any log-normal distribution can be accurately defined by arithmic mean and standard deviation. Actually, there is a one-to-one transformation from the arithmic statistics to the geometric statistics, see e.g. https://en.wikipedia.org/wiki/Lognormal_distribution. My interpretation is that it is not important whether one uses arithmic or geometric statistics, as long as one is aware that their use does not imply (!) either a normal or log-normal distribution.**

We agree here: in fact, our original manuscript cited the table in O'Neill et al (2000) which provides these conversion formulae, to point out the transforms. In the revised manuscript we have changed the wording to make it clear why we are citing the table there, and added additional explicit mentions elsewhere. On the latter point, our experience is that most users take the summary statistic and do not directly consider the underlying distribution (i.e. they are not actively aware of this point). We have also emphasised this point in the revised manuscript.

**Another issue is physical conservation of the property under study. AOD is not a good example so let's consider column burdens of aerosol or trace gases. These may be expected to have log-normal distributions in time and space as well. Describing them with geometric means would cause loss of**

**mass conservation! Consider a dataset at 10 km that is aggregated to 100 km: the arithmic mean preserves total mass in the 100 km grid-box while the geometric mean does not. It seems one has to consider what is causing the log-normality: if it is due to lognormal retrieval errors, geometric means seem justifiable as they ameliorate the effect of outliers. If it is due to the nature of the property, conservation-laws may be more important and arithmic means are to be used. I am sure much more can be said about this.**

**That said, if arithmic means are used to describe log-normal distributions and then carried forward through non-linear analyses under the assumption of normality, significant problems may arise. The authors allude to this on p. 23, l 19 when they talk about parametrisations.**

**It would be great if the authors take the above into consideration when preparing their final manuscript. In all this is a worthwhile discussion.**

Thanks – we partially agree, and these are important points. Our study is one step towards moving to a better treatment of data aggregates. We don't have all the answers so are trying to raise the important points for discussion. The point about mass conservation is an interesting one.

If one needs to estimate total mass and has only one metric to look at, arithmetic mean conserves mass. That is something of a corner case (which might be applicable to studies using AOD to estimate particulate matter levels). In some of those specific applications, a user is likely to want more finely-resolved (spatial and temporal) information, however, so might not even be using data aggregates (and instead go back to level 2 data). However if spatial and/or temporal variation of the parameter is important, the distribution shape becomes key. Further, going beyond one-metric estimates, the combination of (arithmetic or geometric) mean and standard deviation also conserves mass. We mention this in case a reader of the review gets the incorrect impression that lognormal statistics cannot conserve mass.

We have expanded the discussion in the revised manuscript to highlight some more potential applications for which one metric might be more or less useful (previously we had used trends as one and discussed forcing as another), as well as to point out that the observed distribution is a convolution of the true distribution with any measurement/model error.

**Minor comments:**

**p 3, l 33: it would be good to state the relation between arithmic mean (and stddev) with geometric mean (and stddev) for a log-normal distribution. Such relation exists, see https://en.wikipedia.org/wiki/Lognormal_distribution**

As noted above, we'd cited the table in the O'Neill et al (2000) study which provides these conversions. We did not include them in our paper for length reasons. We have made the reason for this reference more explicit in our revised manuscript.

**p 5, l 17: "will overstate the typical level of AOD observed and its variability" . While I understand the authors' intention, it seems to me this sentence suffers from the absence of what is "typical". It would appear that "typical" here refers to the geometric mean as a parameter that defines a log-normal distribution. However, there is a simple 1-on-1 mathematical relation between arithmic and geometric mean of a log-normal distribution. I.e. the arithmic mean defines a log-normal distribution as well as the geometric mean. Hence both arithmic and geometric mean can be used to define what is "typical".**

This is related to the above comment. We have also clarified the use of the word "typical" here in the revised manuscript.

**p 6, l 25: "using Normal-appropriate statistics has systematic quantitative implications for the interpretation of the data." Only if the arithmic mean and stddev are interpreted as defining a normal distribution. It is perfectly possible to calculate both without reference to a normal distribution. Actually, they can define a log-normal as well.**

We agree; note, however, that due to a lack of statistical training many/most users do (implicitly or explicitly) treat these numbers through the lens of Normal statistics. We hope we have articulated this better in the revised manuscript.

**p 6, l3: "these factors may include": I believe turbulence is an important factor in the creation of log-normal distributions?**

Yes (for aerosol size distributions, which will be proportional to AOD if their shape is invariant) – we have mentioned this factor explicitly in the revised manuscript, and reordered these paragraphs. Previously it was implicit in some of the Kok work and textbooks cited. We have expanded the discussion of distributions in nature a bit in the revised manuscript (see also response to a reviewer comment below).

**p 8, l 13: "This quadratic formulation is more robust to calibration problems in individual channels" more robust than what? Maybe consider dropping "more"?**

More robust than the two-channel linear interpolation method which is also used commonly. We have expanded the sentence to say this in the revised manuscript.

**p 11, l 7: "tail-waited" tail-weighted ?**

Thanks; this has been corrected.

**p 12, l 3: Shouldn't arithmic and geometric stddev be compared as well?**

We think the means are more relevant to compare than the standard deviations here, as they are both notionally summary metrics of the same quantity (the "typical" value of the parameter, with attention to the word "typical" as noted in a previous comment). The physical utility of comparing standard deviations is, in our view, less clear, and as most users are using aggregates to get an idea of the AOD is is the means which are of most direct relevance to them. We feel that this four-way categorisation (based on SW test results and differences in the means) is intuitive and sufficient for the purpose at hand, i.e. asking (1) which distribution form is a more appropriate representation and (2) when is is important?

**p 13, l 3: "results for temporal (from AERONET and G5NR) and spatial (from MISR, MODIS, and G5NR) frequency distributions of" This confused me as both Figures show spatial distributions of the WS test. The test, in all cases, was presumably done on time-series of data. The captions to the figures seem to say something different: either data was a temporal aggregate (which suggests G5NR results are at its native resolution) or spatially aggregate (which suggests each 30 min of G5NR data was used). Please clarify this?**

Both Figures 2 and 3 of the original manuscript are spatial distributions. Figure 3 of the original submission is a spatial aggregate (from source level 2/G5NR data), not a temporal aggregate. The others are temporal aggregates. This was discussed in Section 2.2, and is indicated in the caption as well: "aggregated spatially over a day from full resolution to 1°". We have further expanded the captions to Figures 2 and 3 (now 3 and 4) in case of confusion.

**p 14, l 5: "calculating an arithmetic mean when the underlying distribution is Lognormal (or vice-versa) introduces an error smaller than 0.01." I disagree with the use of the word 'error'. No error is incurred at all. It is always possible to calculate arithmic means. Any error is due to limited sample size. See also my previous comments.**

It is an error in the case that what is implied by the statistic is not what is inferred by the data user. We have reworded to "offset" to clarify this in the revised manuscript.

**p 17, l 7: "Note also that the near-universal choice of aggregating daily on a UTC calendar day basis, rather in terms of local solar time, can further complicate matters for locations far from the meridian." For another example, see Schutgens, Partridge & Stier ACP 2016, Fig. 13 & 14.**

Thanks for this reference – we have included it in the revised manuscript (here and elsewhere).

**p 23 , l 11: "Even a small change in reported AOD, if systematic, can have important implications for calculations of climate forcing." But changes (differences) in AOD are far more likely to have a normal distribution.**

We feel that the reviewer here is interpreting "changes" to mean "differences in time" while what we mean is "before you were using this number, now you are using that number". We have reworded this (to "offset") to be clearer in the revised manuscript.

**p 23, l 20: "the same argument may apply if forcing parametrisations are developed from model simulations aggregated in certain ways" This is a fair point. A lot of studies point out the distorting impact of non-linear physics/chemistry when using just the mean to represent a distribution. One example from remote sensing is the plane-parallel bias noted in cloud retrievals of LWP. Note however that such biases exist not because of an arithmetic mean but the representation of any disribution by a single number.**

Thanks – we have expanded the discussion here along these lines (and included the reference in the Short Comment by Y. Chen, as well as another one about rainfall). And yes, this is an instance where one would ideally use the full distribution or at minimum both mean, a measure of width, and an assumed form.

**p 23, l 6: "This implies that no simple scaling correction can be applied to existing data sets to transform between arithmetic and geometric estimates" . Assuming a normal or log-normal distribution, exact transformations exist between mean and stddev of arithmic and geometric statistics.**

We have removed this text (and extended the prior sentence) as we acknowledge it could be misleading. We meant that no scaling is possible unless you know or assume the underlying distribution form.

**Sect 4.2 The analysis in this section seems sound and I have no problems with it. That arithmic means yield different trends than geometric means is no surprise, after all these are different means. However, there is the suggestion that geometric means are better simply because the underlying distribution is log-normal. Rather, geometric statistics make it easier to interpret changes in a log-normal distribution but they do not provide more information (or put differently: the arithmetic statistics are not "wrong"). Note also that trend analysis of changing log-normal distribution really requires geometric stddev to be analysed as well but this is seldom done.**

We agree that arithmetic means here are not "wrong" in a mathematical sense, but rather cause a scientific misinterpretation (because users typically don't consider the distribution something is drawn from). For example if someone reads that AOD changes by X per year then they may expect to go outside on any given day and find AOD lower by X than it was a year ago (roughly speaking). This will not be the case if X was calculated as a trend in arithmetic mean when the underlying distribution is close to Lognormal (because of the relation between arithmetic and geometric mean which the reviewer points out, and because most users do not look at the standard deviation so would not consider this transform). We note that our example trend analysis did use the geometric standard deviation in the estimate of the uncertainty for the regression fit. In the revised manuscript we have added sentences in a few points to clarify that it's not about being "wrong" but about the implicit assumptions an analyst makes.

**p 26, l 6: "but quantitatively have a tendency to overestimate their magnitude." It may be good to repeat here that at these three sites the log-normal distribution is the more appropriate distribution to use (previous analysis, Sect 3). At least that seems to be the suggestion here?**

Yes – at all three sites, the data fell most often into the category 4 (difference > 0.01 and most consistent with draws from a Lognormal distribution). Apologies for the omission of this in the previous version of the paper; we have added the numbers into the revised manuscript.

**p 27, l 20: "estimated trends in geometric mean AOD are smaller in magnitude" Trends in satellite data are over often calculated over regions, not like the point sources the authors have used in their example. I wonder how this will affect these conclusions? At some point the central-limit-theorem should kick in and turn any log-normal distribution into a normal one?**

We believe this is a misinterpretation of the central limit theorem. The central limit theorem implies that when you make multiple estimates of a quantity, those estimates will tend towards a Normal distribution, even if the source data are not Normal (https://en.wikipedia.org/wiki/Central_limit_theorem). In this context, it means that your estimates of the geometric (or arithmetic) mean AOD averaged across the region, if you could make those estimates multiple times, would tend towards a Normal distribution. It does not mean that the underlying AOD field itself becomes closer to a Normal distribution. Indeed, looking at the results in this paper and O'Neill et al (2000), on longer scales it looks like AOD becomes further from, not closer to, a Normal distribution.

After initially submitting this paper we presented the analysis at several venues and a similar comment came up once. As a result we have added some text about the central limit theorem to the paper, including a reference to a review paper dealing in part with this misconception.

**p 27, point 2: this point seems to imply it is ok to average AOD in time and compare satellites with satellites or models, as long as we use the proper mean. The authors know that different sampling of data sources often has a far bigger impact. Maybe it is good to state that here.**

We agree; we mentioned sampling earlier and in the revised manuscript have mentioned again to re-emphasise here. These are all individual pieces of the puzzle.

**p 27, l 9: "root mean square error" This is a difference between two AOD and is likely to be normally distributed.**

It is plausible that RMSE distributions are Normally distributed. What we are saying here is that often in AOD validation exercises, people report RMSE at a given AERONET site (or collection of sites). Yet this is not always a meaningful description of what the level of error is at that location, since it tends to be driven by the high-AOD cases which tend to have higher uncertainty. We have expanded a bullet point here to emphasise for the reader our main point (which was: look at the data and not just the metric).

**Review by Anonymous Referee #3**

**This paper looks at an important yet widely neglected issue in selecting the appropriate summary statistics to create daily or monthly climatology from instantaneous measurements. AOD is used as an example, but the study is applicable to any geophysical variables. The study is well thought out, carefully executed, and clearly presented. I have a few comments below.**

Thank you for the kind words.

**1. It would be appealing to give plausible explanations, from the standpoint of physical mechanisms, to the fact that certain geophysical variables, like AOD, follow lognormal distribution (raindrop size is often described by Gamma distribution, which has a similar skewed shape to lognormal), while some are Gaussian. I find the authors' attempt at P6L5 unconvincing. Statistically, lognormal distribution arises from multiplicative processes while normal distribution is from summation of independent/identically distributed (Gaussian or non-Gaussian) processes (central limit theorem). But I feel that it is hard to relate this statistical interpretation to the physical processes happening in reality. The reason for the skewed distribution of many geophysical variables may be due to the simple fact that they are positively defined. This is also supported by the authors' analysis that in clean conditions (AOD approaching positively zero), geometric and arithmetic means are not all that different.**

Yes, the point about being positive definite is a good one (also made by Adam Povey). We were not trying to convince the reader that AOD must be/is Lognormally distributed, only that there are reasons that it might be (or at the least might not be Normally distributed). We have expanded the discussion of distributions in nature in the revised manuscript (note the Gamma distribution was mentioned already in the context of clouds in the conclusion).

**2. Negative or zero AODs are set to a very small positive value. This has to be done in order to calculate geometric mean. I wonder in such special cases (even if they are rare), does it incur any arbitrary bias that renders use of geometric mean less meaningful compared to arithmetic mean? How would the operational L2-to L3 algorithm deal with the negative or zero retrievals?**

It makes no practical difference since the lowest AODs found tend to be smaller than relevant for many applications. A L2 to L3 algorithm should document if there is any truncation, but ideally L2 algorithms permitting the retrieval of negative AOD should be changed, because this is an unphysical retrieval. AOD of 0 is a thornier case as it is not unphysical, but we'd argue is highly unrealistic. We've added a statement in the revised manuscript to mention this and suggest that any enforced lower limit to avoid negative or zero values is sufficiently low not to bias things.

**3. A possible development for aerosol product is to collate products from different sensors for the overlapping domain (such as the MODIS twins from Terra and Aqua). In that case, which mean is more appropriate, geometric or arithmetic?**

If one believes that the two data sets can be regarded as separate samples from the same population (as might be the case for the two MODIS sensors, with the same design, similar performance, and identical L2 retrieval algorithms), then it would make sense to treat them as one data set when collating their L2 data to make a L3 aggregate. In that sense the evidence of the paper supports Lognormality (geometric mean) as, overall, being a better representation than Normality (arithmetic mean). We're not sure that this question needs a modification to

the manuscript as it is not directly on the main topic of the analysis; in any case, the review and this response will eventually be public and citable via the ACP web page.

**4. The study discussed the impact of geometric vs arithmetic means on trend analysis. How about the aerosol climatic impact? Are L3 products used in climate model evaluation or assimilation that the choice of aggregation method may have important effects?**

We picked trends as one of several possible application areas where it's easy to provide a quantitative example. For evaluating model AOD fields, ideally one should account for sampling differences and (as in the previous comment) geometric mean is probably the best thing to compare if one is only looking at one metric. Note that sampling differences are very important here as well – we cited some references in the paper to highlight this. Of course it would be better to examine the full pdf (or at minimum some measure of average tendency and some measure of variability) but this is not always practical. We did mention aerosol forcing as another aspect but feel that providing too many quantitative examples would lengthen the paper and go out of scope. In response to this and other reviewer comments these aspects are addressed under the generally expanded discussion sections of the paper.